# ELECTRA: A Cartesian Network for 3D Charge Density Prediction with Floating Orbitals

**Jonas Elsborg**[1,2]*† **Luca Thiede**[3,4]*

**Alán Aspuru-Guzik**[3,4,5] **Tejs Vegge**[1,2] **Arghya Bhowmik**[1,2] †

[1]Technical University of Denmark
[2]CAPeX Pioneer Center for Accelerating P2X Materials Discovery
[3]University of Toronto
[4]Vector Institute for Artificial Intelligence
[5]Canadian Institute for Advanced Research (CIFAR)

## Abstract

We present the Electronic Tensor Reconstruction Algorithm (ELECTRA) - an equivariant model for predicting electronic charge densities using floating orbitals. Floating orbitals are a long-standing concept in the quantum chemistry community that promises more compact and accurate representations by placing orbitals freely in space, as opposed to centering all orbitals at the position of atoms. Finding the ideal placement of these orbitals requires extensive domain knowledge, though, which thus far has prevented widespread adoption. We solve this in a data-driven manner by training a Cartesian tensor network to predict the orbital positions along with orbital coefficients. This is made possible through a symmetry-breaking mechanism that is used to learn position displacements with lower symmetry than the input molecule while preserving the rotation equivariance of the charge density itself. Inspired by recent successes of Gaussian Splatting in representing densities in space, we are using Gaussian orbitals and predicting their weights and covariance matrices. Our method achieves a state-of-the-art balance between computational efficiency and predictive accuracy on established benchmarks. Furthermore, ELECTRA is able to lower the compute time required to arrive at converged DFT solutions - initializing calculations using our predicted densities yields an average 50.72 % reduction in self-consistent field (SCF) iterations on unseen molecules.

## 1 Introduction

High-accuracy simulations for the design of materials and molecules at the atomic scale are most often done with density functional theory (DFT) based simulations (Kohn & Sham, 1965), as DFT provides a good balance between cost and accuracy for quantum mechanical simulations of matter (Marzari et al., 2021). However, the O($n^3$) scaling of DFT still limits the system sizes and time scales that can be simulated. Linear scaling ML surrogates, such as neural network potentials trained with a large number of DFT simulations, can alleviate this problem by learning a direct mapping between atomic structure and corresponding energy, forces, and other properties with accuracy similar to those from DFT simulations (Friederich et al., 2021). This approach, although first envisioned three decades ago (Blank et al., 1995), has become successful and popular in recent years based on multiple seminal developments (Behler, 2021; Deringer et al., 2021; Unke et al., 2021c).

---

*These authors contributed equally.

†Corresponding authors: `jels@dtu.dk`, `arbh@dtu.dk`

39th Conference on Neural Information Processing Systems (NeurIPS 2025).

An alternative data-efficient ML-accelerated physics simulation approach can be taken where the underlying fundamental variable of the DFT simulations, the electron density, is predicted directly from atomic structures, without self-consistent field (SCF) iterations. Following the Hohenberg-Kohn theorem (Hohenberg & Kohn, 1964), all ground-state properties can be calculated once this ground-state electron density is known (Grisafi et al., 2022; Bogojeski et al., 2020). In recent years, researchers have addressed this task in multiple ways, differentiated by the representation of density data, the molecular representation, and the ML architecture itself (Grisafi et al., 2018; Chandrasekaran et al., 2019; Jørgensen & Bhowmik, 2022; Rackers et al., 2023). The target electron density is commonly predicted on real space grids (Chandrasekaran et al., 2019; Jørgensen & Bhowmik, 2020; Li et al., 2024) or as an expansion of atom-centered basis functions (Grisafi et al., 2018; Unke et al., 2021a; Cuevas-Zuviría & Pacios, 2021; Rackers et al., 2023) which usually take the form

$$\Phi_{l,m}(\boldsymbol{r}) = R_l(r)Y_{lm}(\theta, \phi), \quad l = 0, ..., L \tag{1}$$

$R_l(r)$ represents the radial dependence relative to a center, while $Y_{lm}(\theta, \phi)$ captures the angular dependencies. Larger quantum numbers $l$ correspond to higher-frequency components. The accuracy of the represented density is dependent on the "quality" of the basis set or the grid density. Two key attributes define the quality of a basis set: The number of basis functions per angular quantum number $l$ and the maximum angular momentum quantum number $L$ included in the expansion.

Different systems and properties necessitate varying levels of basis set complexity. There is no universal basis set that provides both, high accuracy and optimal computational efficiency for all types of systems. Instead, the selection of an appropriate basis set depends on the specific requirements of the system under investigation and requires deep domain expertise.

For example, accurate descriptions of systems involving highly polarizable molecules or those with diffuse electron distributions far away from atom centers may require augmented basis sets like aug-cc-pVTZ (Kendall et al., 1992), which include functions with high angular momentum and diffuse components that have long-tailed radial functions designed for modeling long-range dependencies. For smaller systems or those dominated by core-electron interactions, these basis sets lead to unnecessarily large compute costs. In particular, basis functions with higher angular quantum numbers $L$ incur significant costs.

A more compact representation of densities can be achieved by putting extra basis functions at locations of presumed interest, particularly in areas far away from atoms, with rapidly varying densities. These basis functions are called "floating" orbitals, and their utility is well-established in electronic structure theory (Tao & Pan, 1992; Tao, 1993; Tasi & Császár, 2007). They date back to the floating spherical Gaussian orbital (FSGO) model (Frost, 1968). When chosen wisely, floating orbitals can lead to significant improvements in calculation speed and accuracy (Lorincz & Nagy, 2024) by reducing the need for diffuse and high angular momentum basis functions.

> Well-placed floating orbitals can represent densities more efficiently, using lower maximal angular quantum numbers $L$.
> ELECTRA is the first model to predict floating orbital positions without human input.

However, the optimal locations of floating orbitals are often hard to determine Zheng et al. (2021), and picking good locations therefore requires deep electronic structure domain expertise (Lorincz & Nagy, 2024). Our core contribution is a data-driven solution to this problem. We are training a model that, given a molecular graph, accurately reconstructs ground truth charge densities by predicting the 3D position of floating orbitals as well as the coefficients and parameters that define them.

Since charge densities are rotation invariant, we use a rotation equivariant neural network as the backbone of our model. However, a naive implementation of equivariant neural networks is destined to fail, since good placements of floating orbitals can have lower symmetry than the input molecular graph, as we will discuss in later sections. We address this problem by developing a symmetry-breaking mechanism that retains rotational equivariance. We call the resulting model the Electronic Tensor Reconstruction Algorithm (ELECTRA). We test ELECTRA on the widely used QM9 charge density dataset (Jørgensen & Bhowmik, 2022) and achieve results that are competitive with state-of-the-art while being consistently faster.

## 2 Related work

### 2.1 Charge density prediction

Prior work on machine learning prediction of charge density (CD) generally falls into two main approaches, inspired by earlier non-ML methods. Orbital-based methods are rooted in linear combinations of atom-centered orbitals (LCAO), which take the form

$$\rho(\boldsymbol{r}) = \sum_{i}^{N} \sum_{j}^{N_b^i} \sum_{m=-l_{i,j}}^{l_{i,j}} c_{i,j,m} \Phi_{\alpha_{i,j}, l_{i,j}, m, \boldsymbol{r}_i}(\boldsymbol{r}), \tag{2}$$

where the first sum runs over all atoms, and the other two sum index into all basis functions per atom. $\Phi$ usually takes a form as in 1. In the ML community, methods based on this construction typically predict coefficients $c_{i,j,m}$ extracted from ground truth DFT calculations (Fabrizio et al., 2019; Qiao et al., 2022; Rackers et al., 2023; Cheng & Peng, 2024; del Rio et al., 2023; Febrer et al., 2024) as well as refined radial functions $R_l(r)$ (Fu et al., 2024). This is computationally efficient at inference, and orbital-decomposed density representations can offer enhanced accuracy in describing both total and orbital energies by utilizing flexible, orbital-specific potentials that align closely with many-body spectral properties (Ferretti et al., 2014). However, the fixed choice of basis set often limits representation power unless a large, expensive basis set is used, particularly for complex inter-atomic electronic features. By placing additional orbitals on bond midpoints, (Fu et al., 2024) achieved higher expressivity, albeit at higher computational costs, and the additional requirement of determining bonds. The latter point sounds trivial, but bonds are not always well defined making this difficult.

The second method is inspired by viewing the charge density as a numerical grid (Cerjan, 2013), which must be probed at each point to construct the density. By inserting a graph node that can receive messages from the atomic graph representation (Gong et al., 2019; Jørgensen & Bhowmik, 2022; Koker et al., 2024; Pope & Jacobs, 2024; Li et al., 2024) in each grid point, these models directly predict scalar charge values at grid points, offering high expressiveness and accuracy. Even for small molecules, charge density data contains hundreds of thousands of points, and thus, probe-based models are generally more computationally intensive than orbital-based models.

Once we have the density, two broad strategies emerge:

**Use $\rho(\boldsymbol{r})$ as an initial guess:** Plane-wave based KS-DFT can take $\rho(\boldsymbol{r})$ given on a grid and cheaply map it to plane-wave coefficients using fast Fourier transform (FFT), which allows to resume SCF. This approach was demonstrated by Jørgensen & Bhowmik (2020, 2022). The same idea applies to orbital-free DFT (OF-DFT) Weizsäcker (1935); Mi et al. (2023) which natively operates on grids and allows for much faster, albeit less accurate, SCF and energy predictions.

**Use $\rho(\boldsymbol{r})$ directly:** A potential alternative route is to train on KS-DFT densities and only evaluate the energy with OF-DFT without extra SCF, an approach that is standard for benchmarking OF-DFT functionals and that was shown to give better predictions than self-consistent OF-DFT densities Constantin & Ruzsinszky (2009); Iyengar et al. (2001); Perdew et al. (1988). Other properties that can be evaluated directly without extra SCF are Bader's Theory of Atoms in Molecules, which shows how a topological analysis of $\rho(\boldsymbol{r})$ can be used to define bonds rigorously. This allows us to identify covalent, ionic, and noncovalent interactions Bader & Nguyen-Dang (1981); Bader & Essén (1984); Boto et al. (2017) and construct charge partitioning schemes with widespread use in quantum chemistry. Densities also provide qualitative insights into electrophilic and nucleophilic regions Bader et al. (1984), as well as weaker interactions such as hydrogen bonding Koch & Popelier (1995), and allow us to calculate multipole moments and field tensors. Recently developed usage of $\rho(r)$ includes density-based generative 3D drug design models Ragoza et al. (2022); Wang et al. (2022) and density-based descriptors to study redox processes in electrochemical systems (Laubach et al., 2009; de Blasio et al., 2023; Shang et al., 2022) and electrocatalysts Zheng et al. (2014); Koch et al. (2021). Densities themselves are also often used to build cheap surrogates for observables like ionic diffusion (Kahle et al., 2018).

## 2.2 Equivariance and Cartesian tensors

Many objects in physics transform predictably under symmetry transformations. This property is called equivariance. Formally, a function $\boldsymbol{f} : \boldsymbol{X} \to \boldsymbol{Y}$ is equivariant with respect to a group $\mathcal{G}$ whose elements $\boldsymbol{g} \in \mathcal{G}$ act on $\boldsymbol{X}$ and $\boldsymbol{Y}$, if

$$\boldsymbol{f}(\boldsymbol{g_X}\boldsymbol{x}) = \boldsymbol{g_Y}\mathbf{f}(\boldsymbol{x}) \tag{3}$$

For example, we are often interested in the case where $\mathcal{G}$ is the group of translations, rotations, and reflections, see (Thomas et al., 2018; Geiger & Smidt, 2022; Simeon & De Fabritiis, 2024) for more details. Constructing a machine learning model with the equivariance property (3) provides a strong inductive bias that usually leads to increased data efficiency (Brehmer et al., 2024).

If $\boldsymbol{Y}$ is a function space, $\rho \in \boldsymbol{Y}$, $\boldsymbol{g_Y}$ acts on $\rho$ via the left-regular representation, defined as (Brandstetter et al., 2021):

$$[\boldsymbol{g_Y}\rho](\boldsymbol{r}) = \rho(\boldsymbol{g}^{-1}\boldsymbol{r}) \tag{4}$$

In this work, we are learning a model $\boldsymbol{f}$ that maps from molecular geometry $\mathrm{mol} = \{\boldsymbol{r}_k\}_{k=0}^{N}$ to the electron density

$$[\boldsymbol{f}(\mathrm{mol})](\boldsymbol{r}) = \rho(\boldsymbol{r}|\mathrm{mol}) \tag{5}$$

This mapping is equivariant under rotation $\boldsymbol{R}$

$$\boldsymbol{f}(\boldsymbol{R}\,\mathrm{mol}) = \rho(\boldsymbol{r}|\boldsymbol{R}\,\mathrm{mol}) = \rho(\boldsymbol{R}^{-1}\boldsymbol{r}|\mathrm{mol}) \tag{6}$$

Equivalently, the electron density is rotation invariant under joint rotation of both electron and molecular geometry:

$$\rho(\boldsymbol{R}r|\boldsymbol{R}\,\mathrm{mol}) = \rho(\boldsymbol{R}^{-1}\boldsymbol{R}r|\mathrm{mol}) = \rho(\boldsymbol{r}|\mathrm{mol}) \tag{7}$$

For notational convenience, we will drop the conditioning on mol in our notation from now on and only write $\rho(\boldsymbol{r})$.

Cartesian tensors provide a systematic way to handle rotation equivariance. An $l$th-rank Cartesian tensor $\mathbf{T}$ is an $l$th-rank tensor that transforms under rotation as

$$\boldsymbol{T}_{i_1 i_2 \cdots i_l} \xrightarrow{\boldsymbol{R}} = (\boldsymbol{R}_{i_1 j_1})(\boldsymbol{R}_{i_2 j_2})\cdots(\boldsymbol{R}_{i_l j_l})\boldsymbol{T}_{j_1 j_2 \cdots j_l} \tag{8}$$

where $\mathbf{R}$ is an orthogonal matrix in Cartesian coordinates. Equivariant graph networks can be built to leverage operations on Cartesian tensors (Simeon & De Fabritiis, 2024; Wang et al., 2024) such as linear combinations, tensor contractions, and partial derivatives (Simeon & De Fabritiis, 2024; Wang et al., 2024) that ensure that the network's outputs are equivariant. One example of equivariant networks that operates on Cartesian tensors is the High-order Tensor Passing Potential (HotPP) (Wang et al., 2024). HotPP's node features and messages are arbitrary order Cartesian tensors, and the operations are constrained such that the outputs remain Cartesian tensors. This allows predictions on higher order physical quantities like dipole moments (rank 1 tensors, i.e., vectors) and polarizability tensors (rank 2 tensors, i.e., 3x3 matrices), and similarly allows for more complex atomic environments to be distinguished. In HotPP, an atomistic system is represented as a graph $G = (V, E)$, where $V$ is the set of atoms (nodes) and $E$ is the set of edges (defined up to a cutoff radius) in a molecule. Each atom $A$ is characterized by a feature vector $v_A$, and each edge ($e_{A_1, A_2} \in E$) between atoms $A_1$ and $A_2$ is associated with an edge vector $v_{A_1, A_2}$ and a scalar distance $d_{A_1, A_2}$.

## 3 Methods

### 3.1 The ELECTRA model

For ELECTRA we make a simplified ansatz compared to the LCAO ansatz. Inspired by the idea in Gaussian Splatting (Kerbl et al., 2023) of approximating a complex 3D field with many inexpensive Gaussians, we here use Gaussians as an efficient density basis, because they are closed-form,

separable, and cheap to evaluate on grids (Kerbl et al., 2023; Charlier et al., 2021; Feydy et al., 2020). Concretely, we represent the charge density as a 3D Gaussian mixture model:

$$\rho\left(\mathbf{r}\right) = \text{ReLU}\left(\sum_{A \in M} \sum_{j=0}^{N_A} w_{A,j} \mathcal{N}(\boldsymbol{r}|\boldsymbol{\mu}_{A,j}, \boldsymbol{\Sigma}_{A,j})\right) \tag{9}$$

where ReLU is the standard rectified linear unit function used to prevent negative densities[1], $A \in M$ represents the atoms in the molecule, and $N_A$ is the number of Gaussians for each atom, which can depend on the atom type. The weights $w_{A,j}$ are signed, which improves expressivity. This can, e.g., be used to construct shell-like structures by inserting a negative density at the center of a larger sphere. In the following sections, we elaborate on how all Gaussians are constructed in a per-atom way, using the model output specific to each atom. Gaussians are equivalent to traditional Cartesian orbital functions with angular quantum number $l = 2$, an extra nonlinearity, and simplified radial dependency; see appendix B for why. This is the reason we are saying that ELECTRA uses $l = 2$ orbitals. We will see that, if made "floating", these simplified orbitals are enough to achieve strong performance, in contrast to atom-centered basis functions, which require a high maximum angular quantum number $L$ for good performance. In principle, we could also use more conventional basis functions and make them "floating".

To enforce rotational invariance (7) of the predicted electron density (9), we need the weights $w_{A,j}$ to be rotation invariant, while the means $\boldsymbol{\mu}_{A,j}$ and covariance matrices $\boldsymbol{\Sigma}_{A,j}$ need to be rotation equivariant. In particular, the Gaussian means and covariance matrices need to transform like a *Cartesian* tensor, see appendix C for details.

**Equivariant backbone neural network.** To enforce the constraints on $w_{A,j}$, $\boldsymbol{\mu}_{A,j}$ and $\boldsymbol{\Sigma}_{A,j}$, we use a modified version of the HotPP (Wang et al., 2024) equivariant message-passing network to represent atomistic systems in ELECTRA. We initialize the scalar features as well as the first three rank-1 features in each atom using a tailored embedding function. The initialization is important to the final model and is detailed in the paragraphs below. The graph is updated through a series of HotPP's update layers. We then use the resulting features to predict the parameterization $w_{A,i}$, $\boldsymbol{\mu}_{A,i}$ and $\boldsymbol{\Sigma}_{A,i}$ for the Gaussians in (9) using a readout head layer. Other important changes to the default HotPP implementation are detailed in the paragraphs below, and ELECTRA otherwise follows the reference implementation.

**Atomic embeddings and variable basis set size.** In quantum chemistry, different atoms require differently sized basis sets, since the complexity of the electronic structure generally depends on the atomic number (Weigend & Ahlrichs, 2005) and the number of valence electrons. Inspired by this, ELECTRA predicts a variable number of Gaussians depending on the number of valence electrons. This is achieved by assigning each output channel of each atom in HotPP to one Gaussian. Denoting $M_e$ as the number of Gaussians per valence electron, we can use the first $n_e \cdot M_e$ channels of each atom to represent the Gaussians, where $n_e$ is the number of valence electrons for that atom. For this to work, a channel width of $N_c = 8 \cdot M_e$ is sufficient in HotPP. Each atom $A$ then uses only its first $M_e \cdot n_{e,A}$ channels. For example, oxygen ($n_e = 6$ from the 2s and 2p shells) utilizes $6M_e$ output channels, while hydrogen ($n_e = 1$ from the 1s shell) uses only $M_e$ output channels. We use an atomic embedding function inspired by SpookyNet (Unke et al., 2021b) to represent the atom in terms of its nuclear charge and electron configuration (details are in Appendix G).

**Symmetry-breaking** Since ELECTRA's orbitals are not as expressive as standard spherical harmonics-based orbitals, the model needs to have maximum freedom in placing the orbitals in space. However, equivariant networks prohibit their outputs from having a lower symmetry than their inputs (Smidt et al., 2021; Xie & Smidt, 2024). In particular, *local reflection symmetries* lead to strong restrictions on the outputs of equivariant models. For example, in Figure 1a we plot all vector-valued outputs of a randomly initialized equivariant model with a planar molecule as input. Clearly, the outputs are constrained to the reflection plane, which would severely limit a density constructed based on a Gaussian centered on the predicted locations. See appendix D for a detailed

---

[1]Since the ReLU clamp enforces non-negativity, it could in principle create zero-density plateaus with weak local gradients. In practice, this did not impede training: we trained with and without the ReLU clamp and observed no measurable differences, and thus include it here solely out of principle.

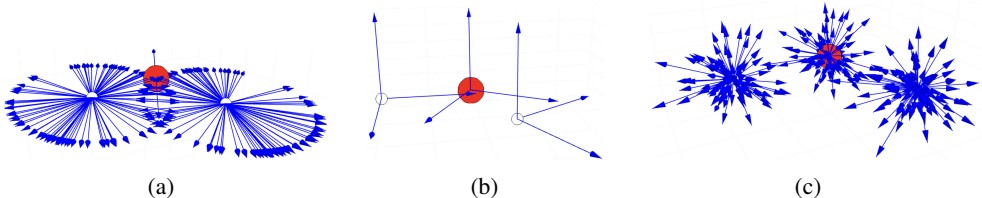

(a)                (b)                (c)

Figure 1: **(a)** Model output without symmetry-breaking: Equivariant neural networks constrain their output to have the same symmetry as the input. If the input molecule is highly symmetric, this leads to highly constrained Gaussian positions. **(b)** To solve this issue, ELECTRA initializes each atom's $l = 1$ vector features with the eigenvectors of the moment of inertia tensor as calculated in that specific atom, which breaks the input symmetry but retains rotational equivariance. **(c)** Model output after first linear layer with symmetry-breaking: The model can learn its own set of symmetry-breaking vectors, allowing output to not be constrained by the symmetry of the input molecule.

mathematical explanation. Since many ground-state geometries are highly symmetric, this poses a big issue for equivariant networks. Previous work has investigated both indirect and direct ways of breaking symmetries. Indirect methods typically relax the equivariance constraints (van der Ouderaa et al., 2022; Kaba & Ravanbakhsh, 2023; Huang et al., 2024), which is undesirable for electron densities since these are exactly equivariant (Rackers et al., 2023). Other methods break symmetries by constructing symmetry-breaking inputs (Liu et al., 2019; Locatello et al., 2020; Xie & Smidt, 2024) or by learning order-breaking parameters during training (Smidt et al., 2021).

> To construct an expressive method for placing floating orbitals, the model must allow for output vectors that belong to a lower symmetry group than the input structures. Thus, a symmetry-breaking mechanism is needed.

Our approach to symmetry-breaking with ELECTRA broadly falls into the category of symmetry-breaking inputs. For our construction, we are first calculating a local moment of inertia (MOI) tensor for each atom:

$$I_{ij}^{(atom)} = \sum_{k=1}^{N} m_k \left( \|\mathbf{r}_k\|^2 \delta_{ij} - x_i^{(k)} x_j^{(k)} \right) \tag{10}$$

Where $k = 1, ..., N$ runs over all atoms inside a local atomic neighborhood defined up to a cutoff radius from the current atom, and the vectors $\mathbf{r_k} = \left( x_1^{(k)}, x_2^{(k)}, x_3^{(k)} \right)$ are calculated relative to the current atom. The three eigenvectors of (10) are also rotation equivariant. Thus, we can use them to initialize the first three $l = 1$ vector features of ELECTRA's GNN, from largest to smallest eigenvalue (Figure 1b), while maintaining rotational equivariance. They define a local coordinate system on which the model can learn its own set of symmetry-breaking objects using a linear embedding layer, the output of which is depicted in Figure1c. Similar ideas of using the moment of inertia vectors have been explored in slightly different contexts in previous work, for example Puny et al. (2021); Duval et al. (2023); Gao & Günnemann (2021); Taniai et al. (2024). The eigenvectors of a matrix are only defined up to a sign flip. One can resolve this issue for example by averaging all possible sign combinations Duval et al. (2023). However, in our case, this averages out all meaningful anisotropies. Instead, we opt for canonicalizing the eigenvectors. Canonicalization of eigenvectors is a research topic that is studied independently (Ma et al., 2024). In this work, we use a sign convention that maximizes the dot product of each eigenvector with the position vector of the center of mass (COM) of the molecule. Mathematically for an eigenvector $\boldsymbol{v}$, we switch the sign according to:

$$\boldsymbol{v}_{\text{canon}} = \begin{cases} \boldsymbol{v}, & \text{if } \boldsymbol{v} \cdot \boldsymbol{r}_{\text{COM}} \geq 0, \\ -\boldsymbol{v}, & \text{if } \boldsymbol{v} \cdot \boldsymbol{r}_{\text{COM}} < 0. \end{cases} \tag{11}$$

There are transformations where this canonicalization will lead to a sign flip, and the model has to learn to compensate for them. However, we find, empirically, that this does not hinder strong performance. There are also some systems in which the moments of inertia degenerate. However, except for some special cases, for example, completely linear molecules, this rarely happens in practice. This is why many successful models rely on the moments of inertia.

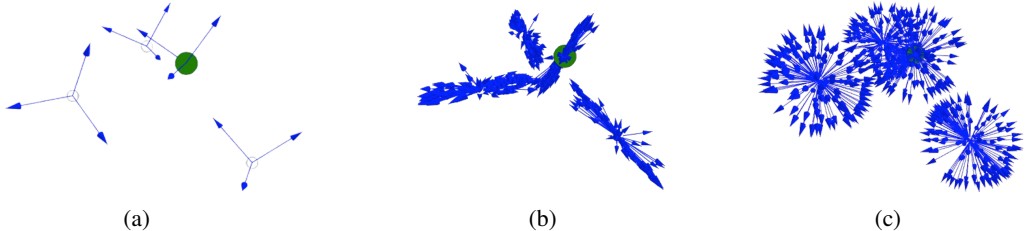

(a)                              (b)                              (c)

Figure 2: (**a**) The initial symmetry-breaking objects of the $NH_3$ molecule (Ammonia). (**b**) The output of a HotPP model with symmetry-breaking but without debiasing layers: The message passing induces a directional bias that concentrates vectors along certain directions, (**c**) The output of our model with debiasing layers: The output vectors don't show any visible bias.

**Debiasing layers.** Even though our symmetry-breaking mechanism allows ELECTRA theoretically to break any symmetry (see appendix D), we observe that the message-passing mechanism of HotPP induces a directional bias of the $l = 1$ features, particularly in highly symmetric molecules. In appendix E, we discuss the origin of this bias mathematically.

For example, in Figure 2b, we plot the output vectors from a randomly initialized HotPP model with the $NH_3$ molecule (Ammonia) and symmetry-breaking objects as input. We see, that the vectors tend to be parallel to the bond axis of the molecule, which is problematic if we were to use these vectors as Gaussian positions because the ground truth density has a lot of density around the bond axis. Empirically, the model was not able to overcome this bias and place the Gaussians efficiently in space, which led to low performance. To address this issue, we are designing a layer that learns to dynamically remove directional biases in the vector features. Our debiasing layer, which we place after every message passing layer, first calculates the covariance matrix of all the $l = 1$ node features associated with each atom:

$$\mathbf{C}_A = \frac{1}{D} \sum_{j=1}^{D} \mathbf{v}_{A,j} \mathbf{v}_{A,j}^T \tag{12}$$

where $\boldsymbol{v}_{A,j}$ are the $l = 1$ features for atom $A$, and $D$ is the channel dimension. We denote $\mathbf{u_{1,A}}$ as the eigenvector of $\boldsymbol{C}_A$ with the largest eigenvalue $\lambda_1$. If there is a directional bias in the features of an atom, $\mathbf{u_1}$ points in the direction of the largest variation. The stronger the directional bias, the larger the magnitude of $\mathbf{u_1}$. We calculate the projection of each $l = 1$ feature onto this principal axis $\mathbf{u_1}$:

$$\mathbf{v}_{A,j}^{\parallel} = (\mathbf{v}_{A,j} \cdot \mathbf{u}_{1,A}) \, \mathbf{u}_{1,A} \tag{13}$$

Note that the sign ambiguity of $\mathbf{u}_{1,A}$ is not important in this case, as $\mathbf{u}_{1,A}$ appears twice in the projection. By subtracting $\mathbf{v}_{A,j}^{\parallel}$ from $\mathbf{v}_{A,j}$ we can reduce the directional bias. To let the model decide how much to subtract, we predict a weight $\mathbf{w}_{A,j}$ using a small neural network, conditioned on $l = 0$ features. The output $\mathbf{w}_{A,j} \in [0, 1]$ is a number that determines how much of the principal direction to remove in each vector, such that the debiased vector is updated as:

$$\mathbf{v}_{A,j} \leftarrow \frac{\mathbf{v}_{A,j} - \mathbf{w}_{A,j} \cdot \hat{\mathbf{v}}_{A,j}^{\parallel}}{||\mathbf{v}_A - \mathbf{w}_{A,j} \cdot \hat{\mathbf{v}}_{A,j}^{\parallel}||}, \tag{14}$$

Where $\hat{\cdot}$ is the dot product. We normalize the vectors in (14) such that the $l = 1$ features only handle directionality, while scales are handled in the readout layer by $l = 0$ predictions. This mechanism, therefore, provides a way to determine and remove bias in the $l = 1$ features.

We note that the symmetry and bias limitations were not unique to the HotPP architecture, and we did initially try e.g. TensorNet (Simeon & De Fabritiis, 2024) and Equiformer v2 (Liao et al., 2023). However, these backbones faced the same issues as HotPP in terms of symmetry-breaking, and ultimately, we opted for HotPP as initial experiments suggested that it worked best.

### 3.2 Density construction

After several message-passing layers, we have a set of features for each atom that we feed into three readout heads. Each readout head produces a set of $l = 0$ (**s**), $l = 1$ (**v**) and $l = 2$ (**M**) features,

$(\mathbf{s_1}, \mathbf{v}_1, \mathbf{M}_1)_{A,j}$, $(\mathbf{s_2}, \mathbf{v}_2, \mathbf{M}_2)_{A,j}$ and $(\mathbf{s_3}, \mathbf{v}_3, \mathbf{M}_3)_{A,j}$, where $A$ indexes the atoms in the molecule, and $j$ the channel. Depending on the atom type, the channel index is $j \in [0, ..., N_e(A) \times M_e]$, where $N_e$ is the number of valence electrons of that atom, and $M_e$ is the number of Gaussians per valence electrons. Intuitively, each of the three heads is specialized on a different distance scale away from the atoms. We use these predictions for the parameterization of the Gaussians in our ansatz (9). Each mean position is calculated as a weighted sum of three $l = 1$ features (one from each head). Similarly, each covariance matrix is a weighted sum of three symmetrized matrices based on the $l = 2$ features. The full details are available in Appendix F.

**Normalization.** Prior work has shown that density prediction models whose output does not integrate to the number of electrons can lead to errors in downstream property predictions (Briling et al., 2021). Thus, as a final step, the densities predicted by ELECTRA are normalized to the number of valence electrons in the system:

$$\rho_{\mathrm{pred}}(\mathbf{r}) = \rho(\mathbf{r}) \times \frac{n_{\mathrm{elec}}}{\int_{\mathbb{R}^3} \rho(\mathbf{r}) \, dV}, \tag{15}$$

where $dV$ represents the differential volume element on the grid. This ensures that

$$\int_{\mathbb{R}^3} |\rho_{\mathrm{pred}}(\boldsymbol{r})| dV = n_{\mathrm{elec}}, \tag{16}$$

Since $n_{\mathrm{elec}}$ is simply the number of valence electrons, this number is already provided as an input to standard DFT codes or can easily be obtained via summation over the valence electrons of each atom.

**Objective function** We train ELECTRA on a loss function $\mathcal{L}$ based on the normalized mean absolute error:

$$\mathcal{L} = \mathrm{NMAE}\,(\rho_{\mathrm{pred}}, \rho_{\mathrm{ref}}) = \frac{\int_{\mathbb{R}^3} |\rho_{\mathrm{ref}}(\boldsymbol{r}) - \rho_{\mathrm{pred}}(\boldsymbol{r})| dV}{\int_{\mathbb{R}^3} |\rho_{\mathrm{ref}}(\boldsymbol{r})| dV} \tag{17}$$

It is not necessary to compute the denominator in (17) during training since the reference grid must integrate to the number of valence electrons, i.e., $\int_{\mathbb{R}^3} |\rho_{\mathrm{ref}}(\boldsymbol{r})| dV = n_{elec}$, and thus the denominator integral can be replaced with $n_{elec}$ during training.

## 4 Experiments

**Dataset and implementation.** We train ELECTRA using the method outlined in Section 3 on reference densities from the QM9 density files. This charge density dataset is the most commonly used benchmark for charge density prediction models. The dataset was generated in VASP (Kresse & Hafner, 1993) using the PBE (Perdew et al., 1996) functional and the Projector-Augmented Wave (PAW) (Blöchl, 1994) method (Jørgensen & Bhowmik, 2022). We use the full split consisting of 123,835 training molecules, 50 validation molecules, and 10,000 test molecules, following the same scheme as in previous related work (Jørgensen & Bhowmik, 2022; Koker et al., 2024; Kim & Ahn, 2024; Cheng & Peng, 2024; Fu et al., 2024). On the QM9 dataset, we train ELECTRA for 864 GPU hours on a single NVIDIA RTX 3090-24GB GPU. We train using a batch size of 1, and do inference on the full grid for both training, validation and test. In comparison to our training protocol, the similar current state-of-the-art method from Fu et al. (2024) was trained for around 1152 GPU hours on NVIDIA A100-80GB GPUs. To further validate ELECTRA, we also perform experiments on the MD dataset consisting of different conformations of six different molecules Bogojeski et al. (2020); Brockherde et al. (2017). A full list of our hyperparameters is given in Table 3 in the Appendix. As an example of how ELECTRA distributes Gaussians around a molecule, we provide Figure 3, which shows the predicted and ground truth densities for $C_7H_9NO$ from the QM9 test set, along with the individual placement of all Gaussians and an error isosurface that visualizes the density errors.

**Benchmark results.** In Table 1 we report the mean accuracy in NMAE [%] (Equation 17) and average inference time per molecule ($t_{inf}$) on the QM9 test dataset of the main ELECTRA model and concurrent charge density prediction models, using results from the original papers as well as results from the model testing carried out in Fu et al. (2024). The SCDP models are the current state-of-the-art, and thus, to ensure a fair comparison, we tested them on our own hardware, i.e., on RTX 3090-24GB GPUs. In Appendix J we report ablated versions of ELECTRA, which demonstrate the influence of the floating orbitals themselves, as well as symmetry-breaking and debiasing layers.

Table 1: QM9 test set performance of various models. **Top**: Results reproduced from Fu et al., 2024 (A100-80GB GPU). †: This version of InfGCN only used Gaussian Type Orbitals (GTOs). **Bottom**: ELECTRA and SCDP (RTX 3090-24GB GPU, this work)

| Metric | i-DeepDFT | e-DeepDFT | GPWNO | InfGCN$^\dagger$ | InfGCN | ChargE3Net |
|---|---|---|---|---|---|---|
| NMAE [%] $\downarrow$ | 0.357 | 0.284 | 0.730 | 3.720 | 0.869 | 0.196 |
| $t_{inf}$ [s] $\downarrow$ | – | – | – | – | 0.833 | 15.18 |

| Metric | ELECTRA (ours) | SCDP ($L=3$) | SCDP+BO ($L=6$) |
|---|---|---|---|
| NMAE [%] $\downarrow$ | **0.176** | 0.432 | 0.178 |
| $t_{inf}$ [s] $\downarrow$ | **0.089** | 0.395 | 1.022 |

(a)      (b)      (c)      (d)

Figure 3: (a) Predicted density for $C_7H_9NO$ using ELECTRA (NMAE = 0.19 %, red = high density, blue = lower density). (b) Ground-truth density. (c) Gaussian placements (red: $w_{A,j} > 0$, blue: $w_{A,j} < 0$). (d) 0.001 $e/\text{bohr}^3$ error isosurfaces (blue = over-prediction, red = under-prediction).

Table 1 shows that ELECTRA is significantly more accurate than the DeepDFT, IncGCN and GPWNO models. Additionally, compared to all timed models in prior publications as well as this work, ELECTRA is faster by at least an order of magnitude. Compared to ChargE3Net, ELECTRA is 0.020 percentage points more accurate (0.176 vs 0.196), and roughly 170 times faster on inference, even when evaluated on inferior hardware (3090-24GB vs A100-80GB). Compared to SCDP, the current SOTA, ELECTRA is about 2.4 times more accurate while being 4.4 times faster compared to the fastest model ($L = 3$), and slightly more accurate than the best model ($L = 6$ + bond-centered orbitals (BO)) while being over 11 times faster. On the MD dataset (Table 2), ELECTRA sets a new state-of-the-art on all six molecules by halving the NMAE % error compared to SCDP. In Figure 4, we also show how ELECTRA distributes the floating orbitals of a Benzene molecule, and how ELECTRA recreates the central density hole of the molecule. These fine details far away from any atom center are hard to model using only atom-centered orbitals. To test how close we are to the optimal solution allowed by our basis, we conduct overfitting experiments (Table 2), for details see H.

**DFT initialization experiment.** Because the iterations of self-consistent field calculations (SCF) dominate wall-clock time in routine electronic-structure DFT workflows, we assessed whether ELECTRA's predicted densities accelerate SCF convergence on unseen molecules. Using VASP with the original dataset settings (Appendix K), we ran paired calculations initialized from (i) atomic superposition and (ii) ELECTRA-predicted densities, recording SCF iterations required to reach convergence at the same tolerance. The results showed that ELECTRA initialization reduces SCF steps by **50.72%** on average. Furthermore, we note that savings track density accuracy: the worst-error molecule ($C_3H_5N_3O_2$, NMAE = 1.43%), which is a significant outlier, shows only 17.4% reduction, whereas the best ($C_9H_2O$, NMAE = 0.153%) achieves a reduction in SCF steps of 60.87%.

> ELECTRA uses floating orbitals and lower-order representations to achieve state-of-the-art density prediction accuracies while being an order of magnitude faster on inference than competing methods. This results in significant reductions in the number of SCF steps required to reach convergence when initializing new DFT calculations with predicted densities.

Table 2: NMAE[%] on MD test sets for ELECTRA, SCDP (Fu et al., 2024), GPWNO (Kim & Ahn, 2024) and InfGCN (Cheng & Peng, 2024), as well as the $N = 1$ Gaussian Overfit experiment. ELECTRA models were trained for 10 GPU hours.

| Molecule | ELECTRA | SCDP | GPWNO | InfGCN | ($N = 1$ Gaussian Overfit) |
|---|---|---|---|---|---|
| MD-ethanol | **1.02** | 2.34 | 4.00 | 8.43 | (0.18) |
| MD-benzene | **0.45** | 1.13 | 2.45 | 5.11 | (0.08) |
| MD-phenol | **0.56** | 1.29 | 2.68 | 5.51 | (0.08) |
| MD-resorcinol | **0.62** | 1.35 | 2.73 | 5.95 | (0.24) |
| MD-ethane | **0.91** | 2.05 | 3.67 | 7.01 | (0.28) |
| MD-malonaldehyde | **0.80** | 2.71 | 5.32 | 10.34 | (0.16) |

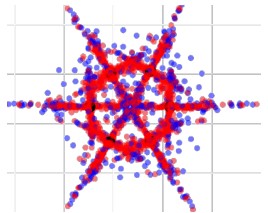

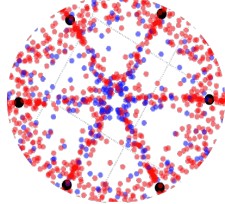

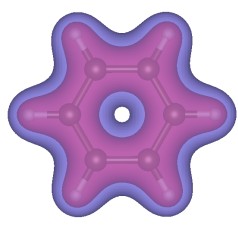

(a) Gaussian centers predicted by ELECTRA for a Benzene molecule in the MD dataset.

(b) *Closeup of (a)*: central density hole of Benzene.

(c) Ground truth density (red = high, blue = low).

Figure 4: Gaussian placements vs. ground-truth electron density for Benzene.

## 5 Discussion

Since no existing basis sets or handcrafted orbital placements are used, ELECTRA is fully data-driven. It represents a shift from a human-designed to a machine-learned density representation, unlike the traditional DFT paradigm, where users must make heuristic element-based choices regarding the basis function type, size, and placement. However, there are still promising opportunities to explore.

**Orbital placement.** Fu et al. (2024) show that adding bond-centered orbitals increases expressivity. For this method to work, bonds must be identified in real-time during training and inference, and may fail for complex systems with non-classical bonding and delocalized interactions, such as partially formed or broken bonds, variable bonding radii, weak interactions like $\pi$-backbonding, and coordination variability. For crystal lattices, issues would arise for, e.g., color centers, where vacancies are occupied by unpaired atomic electrons (Seitz, 1946). Since the vacancy itself does not contain a bond or atom, centered orbitals would likely fail. Similarly, in electrides, the electrons effectively function as anions, requiring non-centered positions (Dye, 2003). In all the above cases, using freely placable orbitals originating from atoms is still viable in theory. ELECTRA would theoretically be able to learn the non-local behavior of the density through (39) or variations thereof. However, this would require specialized methods to replicate floating orbitals periodically. Additionally, floating orbitals scale with the number of atoms rather than the number of bonds, making them computationally more efficient.

**Hybrid Atom-Centered and Floating-Orbital Models.** Naturally, it may be possible to construct floating orbitals using spherical coordinates, which was also suggested in Fu et al. (2024). We thus believe that ELECTRA is complementary to their work and that the benefits of floating orbitals could be combined with the flexibility of the spherical harmonics-based SCDP models. A model with both atom-centered and floating orbitals likely represents the most efficient use of computational resources, since even in ELECTRA, many orbitals are placed around atomic centers (Figures 1a and 3a).

## Acknowledgements

The authors acknowledge financial support from the Independent Research Foundation Denmark, grant number 0217-00326B (DELIGHT), the Novo Nordisk Foundation grant number 87929 (NitroScale), and the Pioneer Center for Accelerating P2X Materials Discovery (CAPeX), DNRF grant number P3.

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

# A Group theory

We provide a short introduction to the relevant group theoretical background, mainly adapted from (Liao et al., 2023; Liao & Smidt, 2022).

**Definition of Groups.** A *group* is an algebraic structure consisting of a set $G$ together with a binary operation $* : G \times G \to G$. A group, satisfies the following four axioms:

1. **Closure:** For all $g, h \in G$, we have $g * h \in G$.

2. **Identity:** There exists an element $e \in G$ such that $g * e = e * g = g$ for all $g \in G$.

3. **Inverse:** For every $g \in G$, there exists an element $g^{-1} \in G$ such that $g * g^{-1} = g^{-1} * g = e$.

4. **Associativity:** For all $g, h, i \in G$, we have $g * (h * i) = (g * h) * i$.

**Symmetry Groups in Three Dimensions.** In this work, we focus on three-dimensional Euclidean symmetry, where the relevant groups are:

1. The Euclidean group in three dimensions $E(3)$: rotations, translations, and inversion in $\mathbb{R}^3$.

2. The special Euclidean group in three dimensions $SE(3)$: rotations and translations in $\mathbb{R}^3$.

3. The orthogonal group in three dimensions $O(3)$: rotations and inversion in $\mathbb{R}^3$.

4. The special orthogonal group in three dimensions $SO(3)$: rotations in $\mathbb{R}^3$.

**Group Representations.** Let $\boldsymbol{X}$ be a vector space. A *representation* of a group $G$ on $X$ is a map

$$\boldsymbol{D_X} : G \to \text{End}(\boldsymbol{X}),$$

where for each $\boldsymbol{g} \in G$, $D_{\boldsymbol{X}}(\boldsymbol{g}) : \boldsymbol{X} \to \boldsymbol{X}$ is a linear transformation. In practice, $D_{\boldsymbol{X}}(\boldsymbol{g})$ is represented as a matrix acting on $\boldsymbol{X}$, so that group actions are matrix multiplications. These representations respect the group structure: for all $\boldsymbol{g}, \boldsymbol{h} \in G$,

$$\boldsymbol{D}(\boldsymbol{g})\boldsymbol{D}(\boldsymbol{h}) = \boldsymbol{D}(\boldsymbol{g} * \boldsymbol{h}).$$

Two representations $\boldsymbol{D}(\boldsymbol{g})$ and $D'(\boldsymbol{g})$ are said to be *equivalent* if there exists an invertible $N \times N$ change-of-basis matrix $P$ such that

$$P^{-1}\boldsymbol{D}(\boldsymbol{g})P = \boldsymbol{D}'(\boldsymbol{g}) \quad \text{for all } \boldsymbol{g} \in G.$$

A representation $D(\boldsymbol{g})$ is called *reducible* if it can be transformed into block-diagonal form, i.e.,

$$P^{-1}\boldsymbol{D}(\boldsymbol{g})P = \begin{bmatrix} \boldsymbol{D}_1(\boldsymbol{g}) & 0 \\ 0 & \boldsymbol{D}_2(\boldsymbol{g}) \end{bmatrix},$$

so that it acts independently on multiple subspaces of $X$. Otherwise, the representation is *irreducible*. Such *irreducible representations* (irreps) are especially useful for building and classifying group representations.

In the case of $SO(3)$, the Wigner $D$-matrices provide a family of irreducible representations. Any representation of $SO(3)$ can be expressed as a direct sum (concatenation) of these irreps:

$$\boldsymbol{D}(\boldsymbol{g}) = P^{-1}\left(\bigoplus_i \boldsymbol{D}^{(l_i)}(\boldsymbol{g})\right)P = P^{-1}\begin{bmatrix} \boldsymbol{D}^{(l_0)}(\boldsymbol{g}) & & \\ & \boldsymbol{D}^{(l_1)}(\boldsymbol{g}) & \\ & & \ddots \end{bmatrix}P, \qquad (18)$$

where $\boldsymbol{D}^{(l)}(\boldsymbol{g})$ denotes the Wigner $\boldsymbol{D}$-matrix of degree $l$.

The degree $l$ can be thought of as frequencies in in a Fourier decomposition; the higher the $l$ the higher frequency.

## B  Gaussians as Cartesian basis functions

The most often found form of basis functions in quantum chemistry is

$$\Phi_{\alpha,l,m}(\boldsymbol{r}) = R_{\alpha,l}(|\boldsymbol{r}|)Y_{lm}(\theta,\phi) \tag{19}$$

where $l$ is the angular quantum number. To build our densities, we contract the basis functions with a set of coefficients $C_{l,m}$, such that the contribution of all basis functions centered at the same spot can be written in Einstein notation as $C_{\alpha,l,m}\Phi_{\alpha,l,m}$

However, a popular alternative in quantum chemistry is Cartesian basis functions. With $\boldsymbol{r} = (x,y,z)$ Cartesian basis functions can be written as:

$$\Phi_{\alpha,n_x,n_y,n_z}(x,y,z) = N(\alpha,n_x,n_y,n_z)R_{\alpha}(|\boldsymbol{r}|)x^{n_x}y^{n_y}z^{n_z} \tag{20}$$

where $N(\alpha,n_x,n_y,n_z)$ is a normalization factor. The angular quantum number in the case of Cartesian basis functions is defined as $l = n_x + n_y + n_z$. Note that there is a strong relation between the angular momentum quantum number for spherical basis functions and Cartesian basis functions. For each $l$, there is an invertible transform between the spherical $l$-shell and the harmonic part of the cartesian $l$-shell (Ribaldone & Desmarais, 2025).
We can collect all the terms above belonging to the same $l$ in one tensor:

$$\boldsymbol{\Phi}_{\alpha,l}(x,y,z) = \boldsymbol{N}(\alpha,l) \cdot R_{\alpha}(|\boldsymbol{r}|)\underbrace{\boldsymbol{r} \otimes \boldsymbol{r} \otimes ... \otimes \boldsymbol{r}}_{l \text{ times}} \tag{21}$$

In particular, for $l = 2$ we get

$$\boldsymbol{\Phi}_{l=2}(x,y,z) = \boldsymbol{N}(\alpha,2) \cdot R_{\alpha}(|\boldsymbol{r}|)\boldsymbol{r}\boldsymbol{r}^{\top} \tag{22}$$

When we contract this with a coefficient matrix $\boldsymbol{C}_{i,j}$, to calculate the contributions of the basis functions to our density, we get

$$C_{i,j}(\boldsymbol{\Phi}_{l=2})_{i,j} = R_{\alpha}(|\boldsymbol{r}|)\boldsymbol{r}^{\top}\underbrace{(\boldsymbol{N}(\alpha,l) \cdot \boldsymbol{C})}_{:=-2\boldsymbol{\Sigma}^{-1}}\boldsymbol{r} \tag{23}$$

If we set the radial term $R_{\alpha}(|\boldsymbol{r}|) = 1$, wrap the remaining term in an exponential function and choose $C$ such that $\boldsymbol{\Sigma}$ is positive definite, we get, up to a normalization constant, a Gaussian:

$$\exp\left(-\frac{1}{2}\boldsymbol{r}^{\top}\boldsymbol{\Sigma}^{-1}\boldsymbol{r}\right) \propto \mathcal{N}(r|0,\boldsymbol{\Sigma}) \tag{24}$$

## C  Equivariance of Gaussians leads to invariant density

The electron density is rotation invariant. In the main text, we claimed that our ansatz (9)

$$\rho(\mathbf{r}) = \sum_{A \in M}\sum_{j=0}^{N_A} w_{A,j}\mathcal{N}(r|\boldsymbol{\mu}_{A,j},\boldsymbol{\Sigma}_{A,j}) \tag{25}$$

is rotation invariant, if the weights $w_{A,j}$ are rotation invariant, and the position $\boldsymbol{\mu}_{A,j}$ and covariance matrices $\boldsymbol{\Sigma}_{A,j}$ transform as $l = 1$ and $l = 2$ Cartesian tensors. It is clear, that the entire ansatz is invariant if each Gaussian is individually invariant. So we need to show, that

$$\mathcal{N}(\boldsymbol{r}|\boldsymbol{\mu},\boldsymbol{\Sigma}) = \mathcal{N}(\boldsymbol{R}\boldsymbol{r}|\boldsymbol{R}\boldsymbol{\mu},\boldsymbol{R}\boldsymbol{\Sigma}\boldsymbol{R}^{\top}) \tag{26}$$

for a rotation matrix $\boldsymbol{R}$. For simplicity, we omit the normalization constant of the Gaussian, since it is rotation invariant. Then we can write

$$\mathcal{N}(\boldsymbol{R}\boldsymbol{r}|\boldsymbol{R}\boldsymbol{\mu},\boldsymbol{R}\boldsymbol{\Sigma}\boldsymbol{R}^{\top}) = \exp\left(-\frac{1}{2}(\boldsymbol{R}(\boldsymbol{r}-\boldsymbol{\mu}))^{\top}(\boldsymbol{R}\boldsymbol{\Sigma}\boldsymbol{R}^{\top})^{-1}\boldsymbol{R}(\boldsymbol{r}-\boldsymbol{\mu})\right) \tag{27}$$

$$= \exp\left(-\frac{1}{2}(\boldsymbol{r}-\boldsymbol{\mu})^{\top}\boldsymbol{R}^{\top}\boldsymbol{R}\boldsymbol{\Sigma}^{-1}\boldsymbol{R}^{\top}\boldsymbol{R}(\boldsymbol{r}-\boldsymbol{\mu})\right) \tag{28}$$

$$= \exp\left(-\frac{1}{2}(\boldsymbol{r}-\boldsymbol{\mu})^{\top}\boldsymbol{\Sigma}^{-1}(\boldsymbol{r}-\boldsymbol{\mu})\right) \tag{29}$$

$$= \mathcal{N}(\boldsymbol{r}|\boldsymbol{\mu},\boldsymbol{\Sigma}) \tag{30}$$

where we have used that $\boldsymbol{R}^{-1} = \boldsymbol{R}^{\top}$. This shows our claim.

# D  Local reflection symmetry constraints

We claim that "local", vide infra, reflection symmetries lead to restrictions on the expressivity of equivariant neural networks. Equivariant neural networks are constrained such that the output transforms with the input:

$$f(\mathbf{g}G) = \mathbf{g}f(G) \tag{31}$$

where $G$ describes our molecules and $\mathbf{g}$ is the action of a group. In particular, we are interested in $\mathbf{g} \in O(3)$, so equivariance to rotations and reflections, in which case we have:

$$f_{l,p}(\mathbf{R}G) = \det(\mathbf{R})^p D_l(\mathbf{R}')f_{l,p}(G) \tag{32}$$

where $D_l(\mathbf{R})$ is the Wigner-D matrix of a rotation, $\mathbf{R}' \cdot \det(\mathbf{R}) = \mathbf{R}$ is the rotation part of an (im)proper rotation and $p$ is the parity of the output, with $p = 1$ called odd and $p = 0$ called even. Next, we define local invariance:

**Definition D.1.** Let $G = \{(r_i - r_j), Z_i, Z_j\}_{i,j=0,...,N-1}$ be the description of a molecule using pairwise displacement vectors and associated atom types. Let $G_i = \{(r_i - r_j), Z_i, Z_j\}_{j \in \mathcal{N}(i,r,L)} \subset G$ be the subset of the graph with nodes within some neighborhood $\mathcal{N}(i, r, L)$ of node $i$, defined by $L$-hop message passing with cutoff distance $r$. Let $\mathbf{R_P}$ be the representation of some reflection symmetry through some arbitrary plane. We call $G_i$ locally symmetric/invariant under $\mathbf{R_P}$ if $\mathbf{R_P}G_i = G_i$.

Importantly, part of a molecule can be locally symmetric, even if the molecule as a whole does not have the same symmetry globally. Take, for example, water in its ground state geometry with O being located at $(0,0,0)$ and $H_1$ and $H_2$ lying in the xz-plane. Then $G_{H_1} = \{(r_{H_1} - r_O, O), (r_{H_1} - r_{H_2}, H_2)\}$ is locally symmetric to reflections along the xz-plane, represented by the matrix

$$\sigma_v = \begin{bmatrix} 1 & 0 & 0 \\ 0 & -1 & 0 \\ 0 & 0 & 1 \end{bmatrix} \tag{33}$$

and we have $\sigma_v G_{H_1} = G_{H_1}$, so $G_{H_1}$ is locally symmetric under $\sigma_v$.

Local symmetry leads to constraints on the outputs of $f(G_i)$. In particular, the output of an equivariant function needs to be an eigenfunction of the symmetry operator $R_P$ with eigenvalue 1:

$$f_{l,p}(\mathbf{R_P}G_i) = f_{l,p}(G_i) = \det(\mathbf{R_P})^p D_l(\mathbf{R_P'})f_{l,p}(G_i) \tag{34}$$

For example, in our water molecule example, we have $R_P = \sigma_v$. For $l = 1$ we have $D_l(\mathbf{R}') = \mathbf{R}'$ and therefore for our $p = 1, l = 1$ we get

$$f_{1,1}(\mathbf{R_P}G_i) = f_{1,1}(G_i) = \det(\mathbf{R_P})\mathbf{R_P'}f_{1,1}(G_i) = \mathbf{R_P}f_{1,1}(G_i) \tag{35}$$

Reflection operators in three dimensions always have exactly one eigenvalue of -1 and two eigenvalues of 1. Therefore, the requirement above restricts the output of the equivariant network to live in the degenerate subspace with eigenvalue 1. For example, the eigenspace of $\sigma_v$ for the eigenvalue 1 is spanned by (1,0,0) and (0,0,1). This means any vector in the xz-plane is an eigenvector of $\sigma_v$. This is exactly what we are seeing in Figure 1a, where all output vectors from the hydrogen atom lie in the xz-plane. It becomes even more restrictive for neighborhoods with more than one reflection symmetry. For example, $G_O$, the descriptor around the Oxygen atom in our water example, has two symmetries. In general, the highest number of symmetries we can have in a molecule is 9 (the full octahedral group $O_h$). However, for most organic molecules it will be at most 6, for example, $CH_4$ with the tetrahedral group $T_d$. If we have more than one reflection symmetry, we have more eigenvalue equations of the form 34, one for each symmetry. Our output has to fulfill all of them simultaneously. This lowers the dimension of the allowed subspace for the output of our equivariant functions by one, since the output needs to live in the union of the allowed subspaces from each reflection individually. For example, if we have both $\sigma_v(xz)$ and $\sigma_v(yz)$ our $l = 1, p = 1$ outputs need to live in the intersection $\text{span}((1,0,0),(0,0,1)) \cap \text{span}((1,0,0),(0,1,0)) = \text{span}((1,0,0))$, which again limits the expressivity of our function. An example of this can be seen on the oxygen atom in Figure 1a.

The discussion shows that local reflection symmetries lead to restrictions on the outputs of equivariant networks. This is a problem since we also need to place Gaussians outside the xz-plane to model the density faithfully.

By augmenting $G_{H_1}$ with a set of vectors $\{s_k\}_{k=0,...,K}$: $G'_{H_1} = G_{H_1} \cup \{s_k\}_{k=0,...,K}$, that are not eigenvectors to $\sigma_v$, we get $\sigma_v G'_{H_1} \neq G'_{H_1}$, and therefore break the symmetry and increase expressivity. Since reflections are never threefold degenerate, such vectors always exist. Given an orthonormal coordinate system as in Figure 1(b), we can learn any vector with a simple linear layer, including $\{s_k\}_{k=0,...,K}$, and therefore break any local reflection symmetry.

## E    Directional bias

As described in the main text 3.1, we observe that message passing in equivariant networks induces a directional bias, see figure 2b. While this is mainly an empirical observation, by examining the message construction in equivariant networks, we get an idea of the origin of this bias. Consider the vector ($l = 1$) valued messages $m^{l=1}_{ij,c}$ from node $j$ to node $i$ with channel dimension $c$, displacement vector between nodes $\boldsymbol{r}_{ij}$, neural network $f_{l_r}(\cdot)$ and node features $h_{l_r,j,c}$:

$$m^{l=1}_{ij,c} = \underbrace{f_{l_r=0}(|\boldsymbol{r}_{ij}|)h_{l_r=0,j,c}}_{\text{Scalars}=:s_c}\boldsymbol{r}_{ij} + \underbrace{f_{l_r=1}(|\boldsymbol{r}_{ij}|)h_{l_r=0,j,c} + \text{Higher body terms}}_{\text{randomly initialized}} \tag{36}$$

$$= s_c \underbrace{\boldsymbol{r}_{ij}}_{\text{observed bias direction}} + \text{random directions}_c \tag{37}$$

So in each layer, we add the displacement vectors while everything else is randomly initialized, leading to the observed bias.

## F    Density construction details

**Scalar factors.**    As a first step, we use the scalar features together with the node embedding $\mathbf{f_j}$ to construct an input for three different MLPs: $\mathbf{s_{inp}}_{A,j} = \begin{bmatrix} \mathbf{f_j}, \mathbf{s_1}_{A,j}, \mathbf{s_2}_{A,j}, \mathbf{s_3}_{A,j} \end{bmatrix}$. The MLPs then predict the Gaussian mixture weights $w_{A,j}$ together with two other sets of scalars to use in the mean and covariance predictions:

$$\begin{aligned}
\mathbf{s_p}_{A,j} &\in \mathbb{R}^3 = \text{MLP}_\text{p}(\mathbf{s_{inp}}_{A,j}), \\
\mathbf{s_m}_{A,j} &\in \mathbb{R}^3 = \text{MLP}_\text{m}(\mathbf{s_{inp}}_{A,j}), \\
w_{A,j} &\in \mathbb{R} = \text{MLP}_\text{w}(\mathbf{s_{inp}}_{A,j}).
\end{aligned} \tag{38}$$

**Gaussian positions.**    ELECTRA places Gaussians (i.e., predicts the mean positions $\boldsymbol{\mu}_{A,j}$) equivariantly by using the $l = 1$ outputs $\mathbf{v}_{1,A,j}, \mathbf{v_2}_{A,j}, \mathbf{v_3}_{A,j}$ and the position scaling factors $\mathbf{s_p}$ of the framework as displacement vectors to the atomic positions:

$$\begin{aligned}
\boldsymbol{\mu}_{A,j} = \mathbf{r}_A &+ \exp\left(\mathbf{s_{p_1}}_{A,j}\right)\mathbf{v_1}_{A,j} \\
&+ \mathbf{s^2_{p_2}}_{A,j}\mathbf{v_2}_{A,j} + \mathbf{s_{p_3}}_{A,j}\mathbf{v_3}_{A,j}
\end{aligned} \tag{39}$$

Therefore, each Gaussian is associated with a position equal to the position $\mathbf{r}_A$ of the atom $A$ it originates from, plus three displacement vectors multiplied by scaling factors. We transform the scaling factors in different ways (exponential, square, and identity) to provide different scales of position displacement, thereby aiming to capture different levels of detail of the output density with each readout head.

**Density prediction using Gaussian mixture models.**    To construct the Gaussian's covariance matrices $\boldsymbol{\Sigma}_{A,j}$ we calculate a weighted sum of the $l = 2$ outputs $\mathbf{M_1}, \mathbf{M_2}, \mathbf{M_3}$. To ensure symmetry and positive semi-definiteness of the covariance matrix, we symmetrize the matrices by constructing the Gram matrices of $\mathbf{M}$, i.e. using the transformation $\mathbf{M} \to \mathbf{M}\mathbf{M}^\top$. For notational simplicity, we omit the $A, j$ subscript for all matrices and scalars in the equations below:

$$\begin{aligned}
\mathbf{s_{G_1}}, \mathbf{s_{G_2}}, \mathbf{s_{G_3}} &= \text{softmax}\left([\mathbf{s_{m_1}}, \mathbf{s_{m_2}}, \mathbf{s_{m_3}}]\right) \\
\boldsymbol{\Sigma} &= \mathbf{s_{G_1}}\frac{\mathbf{M_1}\mathbf{M_1}^\top}{\|\mathbf{M_1}\|_F} + \mathbf{s_{G_2}}\frac{\mathbf{M_2}\mathbf{M_2}^\top}{\|\mathbf{M_2}\|_F} + \mathbf{s_{G_3}}\frac{\mathbf{M_3}\mathbf{M_3}^\top}{\|\mathbf{M_3}\|_F}.
\end{aligned} \tag{40}$$

The Gram matrices are normalized by the Frobenius norm of the original output matrices to preserve the scale for the symmetrized matrices.

## G  Embedding function

To encode information about the electronic and nuclear properties of each atom into the scalar features, ELECTRA uses an embedding function similar to SpookyNet(Unke et al., 2021b), which uses nuclear and electronic embeddings to encode information about the ground state electron configuration of each element. In ELECTRA, each atom's electronic and nuclear properties are encoded as $\mathbf{f} = [P, N, V, E_1, \ldots, E_n, F_1, \ldots, F_n]$, where $P, N$ and $V$ are the numbers of protons, neutrons and valence electrons while $E_i$ and $F_i$ denote orbital occupancies and free-electron counts, respectively, for each orbital. A multi-layer perceptron (MLP) maps $\mathbf{f}$ to the embedding space, $s_{\text{init}} = \text{MLP}_{\text{Emb}}(\mathbf{f})$, which is used as the initial scalar features.

## H  Basis limitation experiment.

Errors are markedly higher on the MD dataset than on QM9 for ELECTRA and baselines. Specifically for ELECTRA, we tested whether this reflects a basis limit of our simplified Gaussian ansatz (Eq. 9). Thus, for each molecule, we overfit an unconstrained Gaussian mixture with the same number of components as ELECTRA, initializing means as atom-center displacements. Using identical loss and optimizer, we trained for 100,000 steps on a single snapshot per molecule. As Table 2 shows, this overfit GMM achieves much lower errors, indicating that the gap is predominantly model error rather than a limitation of the Gaussian basis. Given GMM non-convexity and local minima, these fits are a conservative lower bound on basis expressivity, and we expect that longer or stabilized schedules could do better, and the results are consistent with the theoretical universality of Gaussian mixtures in $L^2(\mathbb{R}^3)$ (Calcaterra & Boldt, 2008).

# I  Hyperparameter table

Summarized in Table 3 below are standard hyperparameters used for all experiments.

Table 3: Hyperparameters for the main ELECTRA model.

| Hyperparameter | Value |
|---|---|
| Graph radius cutoff | 8.0 Å |
| HotPP # body layers | 3 |
| Gaussians per electron ($M_e$) | 90 |
| Graph network channel width | 750 |
| GNN $L_{max}$ - Body layers | 3 |
| GNN $L_{max}$ - Head layers and inference | 2 |
| Body order ($N_{max}$) | 3 |
| Precision | Float32 |
| Optimizer | Adam (Default parameters) (Kingma, 2014) |
| Weight decay | 0 |
| Initial learning rate $LR_{initial}$ | $3.5 \times 10^{-5}$ |
| Learning rate scheduler | Linear ($LR = LR_{initial} \times \gamma^{Epoch}$) |
| Learning rate gamma ($\gamma$) | 0.85 |

# J  Ablation studies

In Table 4 we ablate the main ELECTRA model in five different ways. Not allowing for negative Gaussians ("No negative Gaussians" in the Table) means that the weights $w_{A,j}$ in Equation 9 are restricted to being positive. Removal of scaling ("No scaling") means removing the predicted scaling factors from the displacement vectors in Equation 39 and from the weighted sum of symmetrized covariance matrices in Equation 40. Removal of the debiasing layers is self-explanatory, while removal of the symmetry-breaking mechanism ("No symmetry-breaking") means that we do not initialize the $l = 1$ features of the GNN with the moment of inertia eigenvectors as described in Section 3. Finally, removal of floating orbitals ("No floating orbitals") means that the displacement vectors of Equation 39 are set to zero such that for the the mean position of the j'th Gaussian of the A'th atom, $\mu_{A,j} = r_A$. I.e., all Gaussians become atom-centered.

Table 4: Ablation studies of ELECTRA.

| Model | NMAE [%] ↓ |
|---|---|
| ELECTRA | **0.176** |
| ELECTRA - No negative Gaussians | 2.027 |
| ELECTRA - No scaling | 0.356 |
| ELECTRA - No debiasing layers | 0.584 |
| ELECTRA - No debiasing layers and no symmetry-breaking | 0.705 |
| ELECTRA - No floating orbitals | 6.069 |

# K QM9 DFT initialization experiment parameters

The parameters used to initialize fresh DFT runs with or without an initial density follow the original parameters used by Jørgensen & Bhowmik (2022) to create the QM9 dataset. They are reported in Table 5 below.

Table 5: VASP parameters used to reproduce the DFT initialization experiment on QM9. Baseline uses **ICHARG=2** (atomic superposition). ELECTRA-initialized runs use **ICHARG=1** to read the predicted CHGCAR as the starting density; all other tags are identical.

| Setting (shorthand) | Value |
|---|---|
| Exchange–correlation functional (XC) | PBE |
| Job start mode (ISTART) | 0 |
| Electronic minimization algorithm (ALGO) | Normal |
| Charge density initialization (ICHARG) | 2 |
| Spin treatment (ISPIN) | 1 |
| Max electronic steps (NELM) | 180 |
| Initial non-SCF steps (NELMDL) | 6 |
| Symmetry usage (ISYM) | 0 |
| Correction output (LCORR) | TRUE |
| Ionic time step (POTIM) | 0.1 |
| Min electronic steps (NELMIN) | 5 |
| $k$-point mesh (KPOINTS) | $1\times1\times1$ |
| Smearing method (ISMEAR) | 0 |
| Electronic conv. criterion (EDIFF) | $1.0 \times 10^{-6}$ |
| Smearing width (SIGMA) | 0.1 |
| Ionic steps (NSW) | 0 |
| Subspace diagonalization (LDIAG) | TRUE |
| Real-space projectors (LREAL) | Auto |
| Write wavefunctions (LWAVE) | FALSE |
| Write charge density (LCHARG) | TRUE |
| Plane-wave cutoff (ENCUT) | 400 |

## L  Broader impact

This paper and the ELECTRA model presented in it can dramatically speed up the prediction of charge densities and can potentially be applied to model other 3D densities. Charge density prediction is a foundational problem in computational chemistry, and by accelerating it our approach can advance scientific discovery, though its impact — positive or negative — depends on how it's applied. The greatest benefit is in materials design and drug-discovery workflows that address real-world challenges in e.g. green transition or medicine.

