# OpenReview forum: "ELECTRA: A Cartesian Network for 3D Charge Density Prediction with Floating Orbitals"
_NeurIPS.cc/2025/Conference — NeurIPS 2025 spotlight_

### Official Review · Reviewer_5buA · 2025-06-25

**Clarity:** 3
**Significance:** 3
**Originality:** 3
**Rating:** 5
**Confidence:** 3

**Summary:**

The paper introduces the novel machine learning model ELECTRA for the prediction of the electron density in molecular systems. Contrary to previous approaches, the density is not represented in an atom-centered basis expressed in spherical harmonics, but through floating orbitals using ordinary Gaussian functions. The positions and covariance matrices of these floating orbitals are learned by a neural network alongside the corresponding coefficients. Since the neural network is equivariant ELECTRA also introduces a symmetry breaking mechanism in order to allow for expressive densities also in molecular systems which exhibit a high degree of symmetry. The model achieves state of the art results, while being an order of magnitude faster than models with similar performance.

**Questions:**

- On page 5 it is mentioned that ELECTRA uses an atomic embedding function inspired by SpookyNet, but I could not find any details on the exact implementation. Could the authors provide details on this in the Appendix?
- I would be interested in how well a Gaussian mixture model with the same number of Gaussians as in ELECTRA is able to approximate the target densities in principle in order to know how much of the density error can be attributed to the choice of basis and how much to the limited accuracy of the model.
- If possible I would like to see a comparison to the SCDP+BO (L = 6) model on the MD dataset.

I am willing to increase my score if the above points are adequately addressed.

**Ethical Concerns:**

["NO or VERY MINOR ethics concerns only"]

**Final Justification:**

During the rebuttal the authors addressed all my remaining concerns. I think that exploring alternative density representations besides atom-centered bases in machine learning, is a relevant and promising research topic. This paper makes valuable contributions and presents noteworthy findings in that context.

**Limitations:**

Limitations are addressed. Not being able to directly use the density prediction in conventional Kohn-Sham SCF procedures is another point which might still be added.

**Quality:**

3

**Strengths And Weaknesses:**

- Strengths:
    - The paper is written in a clear and understandable fashion. I also found the blue boxes highlighting key takeaways helpful.
    - The model shows very good results in particular with respect to inference speed at comparable accuracy.
- Weaknesses
    - Although the need for a symmetry breaking mechanism is convincingly demonstrated by the conducted ablation studies, I am not sure about the claim that this is due to local symmetries within the molecules. The mathematical arguments provided in Appendix C are perfectly fine for global symmetries of molecules, but break down for local symmetries in a message passing framework where messages are propagated beyond the locally symmetric regions.  The problem I see is that in the proof it is assumed that the output features of nodes within a sub graph depend only on input features within the same sub graph. For a general message passing network this is however not the case.
    - The final densities are constructed using the ReLU function to ensure positivity everywhere. However this can result in regions of unphysical densities where they are constant 0. This could reduce the performance in downstream tasks, but might also be problematic during training, e.g. when the position of a Gaussian is within such a region of vanishing density. In such a case also the gradient signal on the parameters of the respective Gaussian would become very small.
    - In the comparison of evaluation metrics on the MD dataset, ELECTRA is compared only to the fastest SCDP model, but given the results from the QM9 dataset I would expect the SCDP+BO (L = 6) model to perform much closer (maybe even better) compared to ELECTRA.
    - Since the densities are not represented in a LCAO basis the resulting densities cannot straightforwardly be used as initial guess in conventional Kohn Sham DFT which is one of the most common use cases for machine learned electron densities for molecular systems.

---

> ### Author Rebuttal · Authors · 2025-07-29
>
> We thank the reviewer for their thoughtful and thorough feedback. Below, we address all raised concerns, and provide new results which we hope further reinforces the reviewer's favorable opinion of our work. This includes showing that ELECTRA predictions can indeed be used as initial guesses, as well as a test of the limitations of the choice of basis, as suggested.
>
> >**_Although the need for a symmetry breaking mechanism is convincingly demonstrated by the conducted ablation studies, I am not sure about the claim that this is due to local symmetries within the molecules. The mathematical arguments provided in Appendix C are perfectly fine for global symmetries of molecules, but break down for local symmetries in a message passing framework where messages are propagated beyond the locally symmetric regions. The problem I see is that in the proof it is assumed that the output features of nodes within a sub graph depend only on input features within the same sub graph. For a general message passing network this is however not the case._**
>
> Thank you for taking the time to read our proof in depth! We understand your concern, but the argument still works if we define the symmetric subgraph as the $n$-hop neighborhood of the node, where $n$ is the number of layers. We can upper bound this neighborhood with $n$*cutoff_radius. In that case, no symmetry breaking information from outside the sub-graph would reach the considered node.
> In practice, we did observe problematic behavior even for not completely symmetric graphs, which would need a refined theory to explain fully rigorously. We think our analysis is a good starting point, though: If we perturb a symmetric input graph to be slightly unsymmetric, we know by the smoothness of the neural network that the output would still be close to the completely restricted case. The network would need to learn a very large Lipschitz constant to "fight" these constraints, which is known to be detrimental to model performance.
> Please let us know if this argument makes sense to you, and we will include it in the appendix.
>
> As a side note: Many ground state geometries are also globally symmetric, so even if we disregard local symmetries, the argument is still valid.
>
> >**_The final densities are constructed using the ReLU function to ensure positivity everywhere. However this can result in regions of unphysical densities where they are constant 0. This could reduce the performance in downstream tasks, but might also be problematic during training, e.g. when the position of a Gaussian is within such a region of vanishing density. In such a case also the gradient signal on the parameters of the respective Gaussian would become very small._**
>
> We had the same concern about the training signal and actually tried training both with and without the ReLU, but did not see any difference. We think this is because even if part of the predicted density is zeroed out during training, the Gaussians have support everywhere and therefore still have training signals from all the other points.
> Many other charge density models must contend with the same issue, namely that ML models are not always constrained to predict positive densities, and so choosing whether to use ReLU is essentially choosing whether to risk areas with negative density or areas with zeroed-out densities (as you mention above).
> However, this did not matter much in practice, since even without the ReLU the model learns to never predict negative densities, and we only added the ReLU as a safety mechanism.
>
>
> >**_In the comparison of evaluation metrics on the MD dataset, ELECTRA is compared only to the fastest SCDP model, but given the results from the QM9 dataset I would expect the SCDP+BO (L = 6) model to perform much closer (maybe even better) compared to ELECTRA._**
> >**_If possible I would like to see a comparison to the SCDP+BO (L = 6) model on the MD dataset._**
>
> Unfortunately, we could not find the data preparation scripts for running SCDP on the MD dataset, which makes reproduction difficult. However, the choice of model was made by the SCDP authors themselves, and we merely reference their reported results. Note that the SCDP model used for these experiments is **not** the fastest model. While they use a relatively low K=4 (number of graph layers) and L=3 value (tensor order), they also use an intermediate beta value of beta=1.5 (which determines the basis set size), and importantly, they **do** use the extra bond-centered orbitals and a fine-tuning step. Inclusion of the bond-centered orbitals is the model choice that gives SCDP the largest relative performance gain according to their own experiments (see e.g. Figure 3 in the SCDP paper[1]). As the parameters used are not identical to any set used for the SCDP QM9 experiments it appears that the parameters were optimized by the authors.
>
>
> >**_Since the densities are not represented in a LCAO basis the resulting densities cannot straightforwardly be used as initial guess in conventional Kohn Sham DFT which is one of the most common use cases for machine learned electron densities for molecular systems._**
>
> We actually can use them as an initial guess! A density given on a regular grid can be efficiently transformed into plane wave basis using fast Fourier transformation to initialize SCF cycles. This follows previous work, for example, ChargE3Net [2], and is also how plane wave codes conventionally initialize their calculations from SAD guesses. This is straightforward in practice; in the VASP code base, you can provide custom initializations provided on the grid using the ICHARG=1 setting. (we are not allowed to post links to the documentation, but it can be easily found)
>
> To demonstrate this point, we used the density predictions of ELECTRA to initialize DFT calculations for our QM9 test molecules and achieved an **average reduction of 50.72 % SCF cycles**  compared to conventional initialization. We will add these numbers and a better description how this is done to our paper to demonstrate practicality!
>
>
> >**_On page 5 it is mentioned that ELECTRA uses an atomic embedding function inspired by SpookyNet, but I could not find any details on the exact implementation. Could the authors provide details on this in the Appendix?_**
>
> We will add a small paragraph in the appendix on the model details. We reference here a draft of this paragraph:
>
> *“To encode information about the electronic and nuclear properties of each atom into the scalar features, ELECTRA uses an embedding function similar to SpookyNet. Each atom’s electronic and nuclear properties are encoded as $\mathbf{f}=\left[P, N, V, E_1, \ldots, E_n, F_1, \ldots, F_{\mathrm{n}}\right]$, where $P$, $N$ and $V$ are the numbers of protons, neutrons and valence electrons while $E_{i}$ and $F_{i}$ denote orbital occupancies and free-electron counts, respectively, for each orbital.
> A multi-layer perceptron (MLP) maps $\mathbf{f}$ to the embedding space, $s_{init} = \mathrm{MLP_{Emb}}(\mathbf{f})$, which is used as the initial scalar features. “*
>
> >**_I would be interested in how well a Gaussian mixture model with the same number of Gaussians as in ELECTRA is able to approximate the target densities in principle in order to know how much of the density error can be attributed to the choice of basis and how much to the limited accuracy of the model._**
>
> That is an interesting question!
> We overfitted an unconstrained Gaussian mixture model with means parameterized as displacements from atoms (for better initialization) on a single snapshot from each of the MD molecules with the same number of Gaussians as ELECTRA used. We directly minimized the error using the same loss function and optimizer as ELECTRA for 100,000 steps. The results are reported in the table below. Note that Gaussian mixture models are known to have many local minima; as such the presented result are only a lower bound on the expressivity of the Gaussian mixture model. Longer and more elaborate training (similar to all the tricks employed by Gaussian splatting) would likely result in better accuracy.
>
>
> | Molecule         | ELECTRA | N=1 Gaussian Overfit |
> | ---------------- | ------: | -------------------: |
> | MD-ethanol       |   1.02% |                0.18% |
> | MD-benzene       |   0.45% |                0.08% |
> | MD-phenol        |   0.56% |                0.08% |
> | MD-resorcinol    |   0.62% |                0.24% |
> | MD-ethane        |   0.91% |                0.28% |
> | MD-malonaldehyde |   0.80% |                0.16% |
>
>
> We see that the majority of the error comes from the model, not from the basis choice.
> We will include these experiments in the appendix of our paper
>
> From a theoretical point we also know that Gaussian mixture models are universal approximators in L²(ℝ³) (see e.g. [3] for a proof and discussion of this universal approximation result).
>
> ---
>
> We thank the reviewer for generally acknowledging the strength of our method, and we hope that our responses addressed all the concerns. Please let us know if there are any remaining questions.
>
> ---
>
> [1] Fu X, Rosen A, Bystrom K, Wang R, Musaelian A, Kozinsky B, Smidt T, Jaakkola T. A recipe for charge density prediction. Advances in Neural Information Processing Systems. 2024 Dec 16;37:9727-52.
>
> [2] Koker, Teddy, et al. "Higher-order equivariant neural networks for charge density prediction in materials." npj Computational Materials 10.1 (2024): 161.
>
> [3] Calcaterra, Craig, and Axel Boldt. "Approximating with gaussians." arXiv preprint arXiv:0805.3795 (2008).

---

> > ### Comment · Reviewer_5buA · 2025-08-01
> >
> > I thank the authors for their extensive answers and the new experiments.
> >
> > I think redefining the symmetric subgraph as the n-hop neighborhood makes sense and adds clarity to the argument.
> >
> > Regarding using the output of ELECTRA as initial guess, I referred to the fact that it cannot straightforwardly be used for the relevant class of solvers which require an atom-centered basis. The approach you describe is limited to plane-wave based codes and, as I understand it, also has the additional step of first evaluating the generated density on the grid and performing the Fourier transform, whereas the output of e.g. SCDP can directly be used in suitable Kohn Sham solvers. I still think that this is a relevant limitation for some practical applications that should be mentioned, but not at all an issue which would hinder the publication of the paper.
> >
> > Apart from this I have no remaining concerns. Overall I think that the the paper proposes interesting and novel ideas with intriguing findings, which are worth publishing at NeurIPS. I have raised my score.

---

> ### Author Response · Authors · 2025-08-04
>
> We thank the reviewer for their consideration of our rebuttals and their generally favorable review of our paper.
>
> Regarding the point about the SCDP model as it compares to ELECTRA, we believe it is important to emphasize that none of  the compared density‐prediction methods—including SCDP—can plug its output straight into a gaussian type orbital (GTO) Kohn–Sham code. SCDP does predict coefficients in an atom-centered basis, but for GTO based codes we would require a product basis of atom-centered basis functions, which SCDP (or any of the compared methods) does not use. Thus, one must still (1) evaluate the ML-predicted density on a real-space grid, (2) Fourier-transform (or otherwise project) that grid density into plane waves, and then (3) perform the usual Hamiltonian build, diagonalization, and density‐mixing loops. In other words, both SCDP and ELECTRA serve as a way to provide a near‐converged initial densities for plane-wave codes, but neither approach can be directly plugged into GTO codes.

---

### Official Review · Reviewer_amF4 · 2025-07-01

**Clarity:** 3
**Significance:** 3
**Originality:** 3
**Rating:** 5
**Confidence:** 2

**Summary:**

This paper introduces the Electronic Tensor Reconstruction Algorithm (ELECTRA), which uses floating orbitals to predict electronic charge densities. Inspired by the Gaussian Splatting representation, ELECTRA replaces costly traditional linear combinations of fixed basis functions with floating orbitals. Since designing such orbitals usually requires expert intuition, ELECTRA learns them in a purely data-driven manner, enabling high-resolution charge density prediction. Experiments on well-known benchmarks demonstrate its high accuracy.

**Questions:**

- Why does the method compute directions via eigenvectors? In highly symmetric cases, these eigenvectors are unstable, and noise can easily switch their order or reverse their orientation. How is this handled?
- Debiasing is an important contribution, yet the origin of the bias itself is unclear. Which HotPP component introduces this bias? Understanding the root cause may be more valuable than an ad hoc debiasing fix.
- This approach seems heavily tied to HotPP. While choosing an appropriate backbone is reasonable, what about other equivariant GNNs—or even backbones that ignore symmetry? Since the core idea is to break symmetry within an otherwise symmetric network, broader validation would strengthen the claim.

**Ethical Concerns:**

["NO or VERY MINOR ethics concerns only"]

**Final Justification:**

The authors addressed my concerns with clarity and rigor. I support the acceptance of the paper.

**Limitations:**

yes

**Paper Formatting Concerns:**

No concerns.

**Quality:**

3

**Strengths And Weaknesses:**

Strengths:
- The narrative is clearly explained. The paper introduces floating orbitals and justifies their data-driven prediction via machine learning, which makes the paper easy to follow.
- The authors pinpoint the directional bias that arises in HotPP message passing and propose a dedicated Debiasing Layer to address it.
- Experimental results show that ELECTRA achieves fast and accurate charge-density prediction.

Weaknesses:
- The model largely inherits purely the equivariance property from the existing HotPP framework.
- L.224: "However, we find, empirically, that this does not hinder strong performance." Since equivariance is essential to the method, substantial empirical or theoretical justification is necessary.
- A key contribution of Gaussian Splatting is its tight link to efficient, differentiable rendering and optimization. However, ELECTRA does not exploit this property; it merely outputs a set of Gaussians. Alternative representations, such as point clouds or sparse voxels, should be considered. Alternatively, a loss function tailored to the Gaussian representation must be provided to justify this design choice.

---

> ### Author Rebuttal · Authors · 2025-07-29
>
> We thank the reviewer for thoughtful critique and clear summary of our contributions. While ELECTRA builds on the HotPP framework, its floating-orbital parameterization, symmetry-breaking initialization, and debiasing layers are what sets it apart from standard applications of equivariant models. Below, we address the concerns regarding these design choices and clarify any other issues.
>
> >**_The model largely inherits purely the equivariance property from the existing HotPP framework._**
>
> It is very common for authors to repurpose backbone architectures for different tasks. Most previously published competing methods did so as well, most recently SCDP which was published in last year's NeurIPS proceedings[1]. For our purpose, the HotPP model is well-suited since it outputs Cartesian tensors of varying ranks, which have exactly the properties needed to construct our densities. Could you expand on why this is interpreted as a weakness of our method?
>
> >**_L.224: "However, we find, empirically, that this does not hinder strong performance." Since equivariance is essential to the method, substantial empirical or theoretical justification is necessary._**
>
> The empirical justification is provided in the form of the results themselves, which are state-of-the-art on both the QM9 dataset and the MD dataset. The MD dataset is specifically designed for testing model performance on molecular dynamics (MD) snapshots of the same molecules. These snapshots span changes in bond stretches, angle bends and torsional motions[2], and thus presents a benchmark where any weakness due to the above-mentioned assumption would be exposed. However, the results show the contrary; we see the most significant accuracy gap to the prior state-of-the-art on this dataset, and we achieve more than 2x better accuracy on all MD molecules.
>
> If you have a specific experiment in mind that we should run we can try to do so.
>
> >**_Alternative representations, such as point clouds or sparse voxels, should be considered._**
>
> Thank you for the suggestion, however we are not aware of any work that represent charge densities as point clouds or sparse voxels. If you have papers in mind that we should be comparing to please let us know the references and we will do so.
>
> There are some theoretical reason to believe that these approaches would not work well though: Point clouds can not naturally represent a function $\rho(\vec{r}): ℝ^3 \rightarrow ℝ$. If you mean to encode the atoms with point cloud methods we are unsure how this will be different from the equivariant network. Voxel based methods cannot be built $SO(3)$ equivariant as far as we know, which was shown by many works to be highly beneficial for molecular property prediction. Achieving the sub angstrom resolution neccessary would also be very expensive.
>
> >**_A key contribution of Gaussian Splatting is its tight link to efficient, differentiable rendering and optimization. However, ELECTRA does not exploit this property; it merely outputs a set of Gaussians. Alternatively, a loss function tailored to the Gaussian representation must be provided to justify this design choice._**
>
> The justification for the choice of Gaussians is that their simplicity allows for very efficient evaluation, as shown by the order of magnitude speedup compared to previous approaches, analogously to the reasoning for Gaussians in Gaussian splatting.
> While we informally mention in the paper that our method is inspired by Gaussian Splatting, this is mainly in the sense of placing a large number of cheap-to-evaluate Gaussians in space to approximate an intricate 3D field. This is also why we did not explicitly call our method Gaussian splatting.
> However, regardless of the chosen nomenclature for our approach, the method is completely novel in the field of ML-driven density prediction, while achieving state-of-the-art accuracy and speed.
> However, if you have ideas for how the gaussian structure could be further exploited, we would be happy and interested to pursue such directions!
>
>
> >**_Why does the method compute directions via eigenvectors? In highly symmetric cases, these eigenvectors are unstable, and noise can easily switch their order or reverse their orientation. How is this handled?_**
>
> We are using eigenvectors in two parts of the model; for the symmetry breaking and the debiasing module. We will therefore answer this question in two parts:
>
> For symmetry breaking:
> Our goal is to learn symmetry-breaking, equivariant inputs. For this we need an equivariant local coordinate system. The moment of inertia provide exactly this equivariant orthogonal coordinate system and are for that reason widely used in molecular modeling, for example in stochastic frame averaging [3] and for equivariant wave functions [4]. These methods either learn to become robust to sign flips [3] or use extra canonicalizaton [4]. We choose to canonicalize, see equation 8. While this might not be mathematically elegant, as our results and also the cited methods show, it is a pragmatic solution that does work well in practice.
>
> For debiasing:
> The eigenvectors of the covariance matrix are the direction with the largest variances. By subtracting the largest eigenvector we can therefore learn to remove the direction with the most variance, which corresponds to the bias direction.
>
> The covariance matrix is built from the learned vectors.
> Both experimentally (see figure 2b) and due to the mathematical argument we provide for the next question below, we know that there is always a biased direction. The largest eigenvector will therefore never be degenerate and therefore not unstable to noise.
> We also do not see any instabilities either during training nor validation on either of the datasets.
>
> Orientation flips are no problem since the eigenvector appears twice, see line 245.
>
>
> >**_Debiasing is an important contribution, yet the origin of the bias itself is unclear. Which HotPP component introduces this bias? Understanding the root cause may be more valuable than an ad hoc debiasing fix._**
>
> Since submission we have identified the mathematical origin of the bias:
> Consider the vector ($l=1$) valued messages $m_{ij,c}^{l=1}$ from node $j$ to $i$ with channel dimension $c$, displacement vector between nodes $\vec{r}_{ij}$, neural network $f\_{l\_r}()$ and node features $h\_{l\_r,j,c}$:
>
> $$
> \begin{aligned}
> m\_{ij,c}^{l=1} &= \underbrace{f\_{l\_r=0}(|r\_{ij}|) h\_{l\_r=0,j,c}}\_{\text{Scalars} =: s_c} \vec{r}\_{ij} + \underbrace{f\_{l\_r=1}(|r\_{ij}|) h\_{l\_r=1,j,c} + \text{Higher body terms}}\_{\text{randomly initialized}} \\\\
> &= s\_c \underbrace{\vec{r}\_{ij}}\_{\text{Observed bias direction}} + \text{random directions}\_c
> \end{aligned}
> $$
> So in each layer we add the displacement vectors while everything else is randomly initialized, leading to the observed bias. We will add this explanation in the appendix.
> We cannot just set $s=0$ and ignore the displacement vector since this vector is what carries the geometric information of the molecule. Therefore we still think that our debiasing solution is very sensible.
> Note also that other equivariant networks like TensorNet [2], Equiformer [5], MACE [6] etc. construct their messages in similar ways and would therefore lead to the same bias.
> Please let us know if this explanation makes sense to you and we will include it in the paper.
>
>
> >**_This approach seems heavily tied to HotPP. While choosing an appropriate backbone is reasonable, what about other equivariant GNNs—or even backbones that ignore symmetry? Since the core idea is to break symmetry within an otherwise symmetric network, broader validation would strengthen the claim._**
>
> The approach is not particularly tied to HotPP, as the main differentiating factor for ELECTRA is rather the way it interprets the Cartesian tensor outputs as the building blocks of a Gaussian-based density, as well as the necessary changes that make the task more tractable (symmetry-breaking, debiasing, etc.). This could in principle be done with other models, and we did initially try e.g. TensorNet [2] and Equiformer v2 [5] . However, these backbones faced the same issues as HotPP in terms of symmetry-breaking, and in the end we opted for HotPP as initial experiments suggested that it worked best. We will mention this in the paper.
>
>
> ---
>
> We hope we were able to address all concerns. Please let us know if there are open questions that could be addressed to recommend our paper for acceptance.
>
> ---
>
> [1] Fu, Xiang, Andrew Rosen, Kyle Bystrom, Rui Wang, Albert Musaelian, Boris Kozinsky, Tess Smidt, and Tommi Jaakkola. "A recipe for charge density prediction." Advances in Neural Information Processing Systems 37 (2024): 9727-9752.
>
> [2] Simeon, Guillem, and Gianni De Fabritiis. "Tensornet: Cartesian tensor representations for efficient learning of molecular potentials." Advances in Neural Information Processing Systems 36 (2023): 37334-37353.
>
> [3] Duval, Alexandre Agm, et al. "Faenet: Frame averaging equivariant gnn for materials modeling." International Conference on Machine Learning. PMLR, 2023.
>
> [4] Gao, Nicholas, and Stephan Günnemann. "Ab-initio potential energy surfaces by pairing GNNs with neural wave functions." arXiv preprint arXiv:2110.05064 (2021).
>
> [5] Liao, Yi-Lun, et al. "Equiformerv2: Improved equivariant transformer for scaling to higher-degree representations." arXiv preprint arXiv:2306.12059 (2023).
>
>
> [6] Batatia, Ilyes, et al. "MACE: Higher order equivariant message passing neural networks for fast and accurate force fields." Advances in neural information processing systems 35 (2022): 11423-11436.

---

> > ### Author Response · Authors · 2025-08-05
> >
> > Several other reviewers raised the question of whether our predictions can be used to initialize DFT calculations in practice.
> >
> > To demonstrate that they can, we used the density predictions of ELECTRA to initialize DFT calculations for the QM9 test molecules and achieved an average reduction of **50.72 % SCF cycles** compared to conventional initialization with the same DFT settings used for the dataset creation. We will include this and the precise DFT settings in our revision.
> >
> > We thought that this might be of interest to you as well.

---

> > ### Comment · Reviewer_amF4 · 2025-08-06
> >
> > Thank you, and I apologize for the delay in response.
> >
> > After reading the authors' rebuttal and the discussion with other reviewers, I would like to offer the following remarks and clarifications regarding my initial review.
> >
> > >> The model largely inherits purely the equivariance property from the existing HotPP framework.
> >
> > > It is very common ...
> >
> > Regarding my earlier comment that the model "largely inherits the equivariance property from the existing HotPP framework," I now recognize that this was a mischaracterization. Utilizing an existing backbone is neither a strength nor a weakness in itself. It is a common and reasonable design choice in model development. I appreciate the clarification and apologize for this misstatement.
> >
> >
> >
> > >> L.224: "However, we find, empirically, that this does not hinder strong performance." Since equivariance is essential to the method, substantial empirical or theoretical justification is necessary.
> >
> > > The empirical justification is ...
> >
> > I still believe it would be helpful to include a targeted ablation study to specifically isolate the impact of equivariance-breaking or eigenvector instabilities. For instance, introducing random sign flips in the eigenvector-based coordinate system could serve as an informative test. This would complement the current results and strengthen the theoretical claim with a direct empirical counterpart.
> >
> >
> >
> > >> Alternative representations, such as point clouds or sparse voxels, should be considered.
> >
> > > Thank you for the suggestion, however ...
> >
> > Thank you for the detailed response regarding point cloud and voxel-based representations. I now understand.
> >
> >
> >
> > >> A key contribution of Gaussian Splatting is its tight link to efficient, differentiable rendering and optimization. However, ELECTRA does not exploit this property; it merely outputs a set of Gaussians. Alternatively, a loss function tailored to the Gaussian representation must be provided to justify this design choice.
> >
> > > The justification for the choice of Gaussians is ...
> >
> > I also understand it. This paper's approach can be more accurately described as a GMM-like spatial function approximation using learned Gaussians by my perspective.
> >
> >
> >
> > >> Why does the method compute directions via eigenvectors? In highly symmetric cases, these eigenvectors are unstable, and noise can easily switch their order or reverse their orientation. How is this handled?
> >
> > > We are using eigenvectors in two parts of the model...
> >
> > The explanation for how eigenvectors are used in both symmetry breaking and debiasing is clear and convincing. I particularly appreciate the discussion of canonicalization and the argument that the principal bias direction remains stable. However, if possible, I would still recommend complementing Figure 2 with some statistical analysis or histogram-based evaluation over multiple examples to quantify the typical magnitude and consistency of the bias.
> >
> >
> >
> > >> Debiasing is an important contribution, yet the origin of the bias itself is unclear. Which HotPP component introduces this bias? Understanding the root cause may be more valuable than an ad hoc debiasing fix.
> >
> > > Since submission we have identified ...
> >
> > I consider this analysis to be a meaningful contribution on its own. I strongly encourage the authors to include this explanation in the final version of the paper. Moreover, showing that similar biases may arise in other equivariant networks further elevates the relevance of the proposed debiasing technique.
> >
> >
> >
> > >> This approach seems heavily tied to HotPP. While choosing an appropriate backbone is reasonable, what about other equivariant GNNs—or even backbones that ignore symmetry? Since the core idea is to break symmetry within an otherwise symmetric network, broader validation would strengthen the claim.
> >
> > > The approach is not particularly tied to HotPP, ...
> >
> > It is good to know that other backbones were considered, and I believe including even brief experimental results or discussion in the paper would further support the generality of your design choices.
> >
> >
> >
> > The authors have addressed my concerns with clarity and thoroughness. I am now convinced of the method's contribution, particularly in the context of accurate and efficient electronic density modeling using equivariant networks. I thank the authors again for their careful responses and encourage acceptance of the paper.

---

### Official Review · Reviewer_9wy9 · 2025-07-02

**Clarity:** 2
**Significance:** 3
**Originality:** 3
**Rating:** 4
**Confidence:** 4

**Summary:**

This work proposes predicting electronic charge density using floating orbitals inspired by Gaussian splatting. By leveraging an equivariant neural network architecture, symmetry-breaking initialization, and debiasing layers, the model aims to predict charge densities directly from molecular structures.

**Questions:**

1.	Interpretation of Figures 1 and 2: What is the frame or vector space represented in these figures? Specifically, are the vectors shown in Figures 1a and 1c the predicted mean positions $\boldsymbol{\mu}_{A,j}$ in Equation (6)? A detailed explanation of what each vector represents in these figures would greatly improve clarity.
2.	GPU Memory Requirements: How much GPU memory was used during training and inference? Are there specific trade-offs in inference performance compared to prior work?
3.	Channel Dimension Limitation: Line 184 specifies $N_c = 8 \cdot M_e$. Does this imply that the model can only handle atoms with fewer than 8 valence electrons? How generalizable is the model to elements beyond the first two rows of the periodic table?
4.	Density Grid Sampling: How did you handle sampling of the density during training? Did you employ specific sampling algorithms over the reference density grids in the dataset?
5.	Practical Use Cases: Could you elaborate on whether ELECTRA’s predicted electron densities can improve computational physics simulations, such as providing initial guesses for SCF iterations or being directly used in orbital-free DFT calculations?

---
update Quality 1 -> 2, Clarity 1 -> 2

**Ethical Concerns:**

["NO or VERY MINOR ethics concerns only"]

**Final Justification:**

First, I would like to commend the results on SCF cycle reduction—this is a remarkable contribution that significantly enhances the practical value of the work. I also appreciate the overall improvement in density estimation achieved by the proposed framework.

However, I would like to clarify that some concepts are misused in the manuscript. Specifically, I find the use of the term invariant in the context of electron density somewhat awkward—even when conditioned on the molecular configuration. In group theory, invariance under a group action is typically defined as $f(Rx) = f(x)$. Given that the motivation and architecture of your work are grounded in handling symmetries, I recommend using group-theoretic terminology more deliberately and precisely. A more rigorous explanation—supported by clear definitions and formal justifications—of how symmetry is handled in your framework would substantially strengthen the paper.

Regarding the use of Cartesian tensors, I appreciate the references you provided, which helped me better understand your intentions. However, I still feel that further explanation of how Cartesian tensors are used for representing electron density is needed in the manuscript. Additionally, there appears to be some confusion between geometrical tensor rank (ex. l=1 for force, l=2 for stress 3x3 matrice) and angular momentum quantum numbers, particularly in Equation (5) and lines 140–141. While I agree that using CGTOs (Cartesian Gaussian Type Orbital) makes sense, the proposed model “HotPP” does not appear to be a Cartesian tensor in the conventional sense, but rather a tensor used in mathematical physics—often referred to as a geometrical tensor. I strongly encourage the authors to improve the consistency and rigor of their mathematical formulations. Despite the strong empirical performance, such conceptual imprecisions may mislead readers and weaken the scientific impact of the work.

Finally, regarding the reported reduction in SCF steps, to the best of my knowledge, this metric is sensitive to various hyperparameter choices—such as the functional, pseudopotentials, and the integration grid level. I hope that these choices were adequately and fairly reported in future manuscripts.

Other aspects of the rebuttal addressed my concerns well. While I am hesitant to recommend outright acceptance at this stage, I believe this work is both timely and promising. With improved **mathematical rigor around equivariance**, **clearer distinction between geometric tensors and Cartesian basis functions**, **consistent notation**, and **careful correction of minor issues and typos**, this paper has the potential to make a significant contribution to the ML community. Accordingly, I have raised my score.

**Limitations:**

yes

**Quality:**

2

**Strengths And Weaknesses:**

Strengths:
- Utilizing a Gaussian splatting-inspired idea for electron density prediction is clever and leads to superior performance and significantly reduced inference time.

Weaknesses:
- The manuscript makes a major assumption regarding the rotational invariance of the electron density, which is conceptually flawed.
  - If the charge density were strictly rotationally invariant, then $\rho(\mathbf{r}_1) = \rho(\mathbf{r}_2)$ would hold for all vectors with the same magnitude, i.e., $\|\mathbf{r}_1\|=\|\mathbf{r}_2\|$. This is clearly not the case for most molecular charge densities.
  - In my view, the authors appear to confuse invariance of the physical quantity under rigid-body rotation of the molecule with the concept of group action on function spaces (regular representations). For example, as in [1] (see Equation 2), a rotated density transforms as $\rho(\mathbf{R} \mathbf{r}) = (T_{\mathbf{R}} \cdot \rho)(\mathbf{r})$, which is not invariance.
- A more detailed introduction to Cartesian tensors (e.g., covariances) is necessary, especially given that the topic lies at the intersection of quantum chemistry and machine learning. Many ML researchers may not be familiar with the geometric tensor concepts; the background and appendices do not sufficiently support understanding.
- While using Cartesian tensor-based basis functions is an interesting idea, the manuscript does not provide adequate justification for why decomposing functions in this way is natural. Additional explanations comparing conventional basis choices—such as Gaussian-type orbitals, Slater-type orbitals, or plane-wave basis functions—to Cartesian basis functions would greatly aid comprehension.
- The statement that “Gaussians are equivalent to traditional Cartesian orbital functions with angular quantum number $l=2$, an extra nonlinearity, and simplified radial dependency” is unconvincing. The relationship between spherical harmonics-based basis functions and Cartesian functions is not clearly described. Moreover, while the manuscript refers to higher-order tensors (e.g., vectors for dipoles or matrices for polarizabilities), the connection between geometric tensor rank and quantum angular momentum is fundamentally different and requires clarification.
  - Specifically, the relationship between Equation (15) and Equation (16) is ambiguous. There is no explanation of why the angular quantum number $l$ in both equations should be interpreted identically.


Typos and Minor Issues:
- Equation (2): $\mathbf{r}_i$ is not defined.
- Inconsistent notation between \mathbf{} and \boldsymbol{} for vectors, e.g., $\mathbf{f}$ vs. $\boldsymbol{f}$, and $\mathbf{v}$ vs. $\boldsymbol{v}$.
- Duplicated results for InfGCN in Table 1.
- Line 192: Inconsistent use of “figure” and “fig”; consistent referencing with \ref or \Cref in LaTeX is recommended.
- Equation (11): The definition of $\hat{\cdot}$ is missing.
- Line 208: The notation k=1..N is informal; it should be $k=1,\ldots,N$.
- Line 538: Inconsistent use of $\top$ and $T$ for transposition.
- Line 555: Potential typo, where $g$ should be $G$.
- Equation (18): $\Phi_{l=2}$ should be $\Phi_{\alpha, l=2}$ for clarity.

---

> ### Comment · Reviewer_9wy9 · 2025-07-25
> **Omitted reference**
>
> I apologize for omitting the reference earlier — I have now attached it below.
>
> [1]: Bekkers, Erik J., et al. "Roto-translation covariant convolutional networks for medical image analysis." Medical Image Computing and Computer Assisted Intervention–MICCAI 2018: 21st International Conference, Granada, Spain, September 16-20, 2018, Proceedings, Part I. Springer International Publishing, 2018.
>
> What I intended to convey was that the concept of the regular representation, and it is a fundamental and widely used notion in the fields of group theory and equivariance. And it is about how functions transform under group actions.

---

> ### Author Rebuttal · Authors · 2025-07-29
>
> We thank the reviewer for the thorough review. We believe a notational oversight may have given the wrong impression about symmetry properties, which led to the primary concern raised. We address this and all other concerns below.
>
> >**_"The manuscript makes a major assumption regarding the rotational invariance of the electron density, which is conceptually flawed.
>     - If the charge density were strictly rotationally invariant..."_**
>
> We apologize for the confusion. You are right that there was an error in our notation in equation 4. It should have been
> $$\rho(Rr | \{R r_\text{nuclei,1},..., R r_\text{nuclei,N}\}) = \rho(r | \{r_\text{nuclei,1},...,r_\text{nuclei,N}\})
> $$
> In other words, the electron density is invariant under simultaneous rotation of the molecule and electron/point of evaluation.
>
> Thus, we want to stress that this is not a conceptual flaw, but simply a **notation mistake**.
>
> The construction of our model encodes the correct symmetries both in theory and in practice. This is also evidenced by the state-of-the-art accuracy of our model, which would not have been possible if we had assumed the wrong transformation properties and thereby erroneously overconstrained the model.
>
> >**_A more detailed introduction to Cartesian tensors (e.g., covariances) is necessary, ..._**
>
> Thank you for the feedback. We will expand on the background knowledge in the appendix. Due to the character limit, we unfortunately can't post the section here.
>
>
> >**_While using Cartesian tensor-based basis functions is an interesting idea, the manuscript does not provide adequate justification for why decomposing functions in this way is natural. Additional explanations comparing conventional basis choices—such as Gaussian-type orbitals, Slater-type orbitals, or plane-wave basis functions—to Cartesian basis functions would greatly aid comprehension._**
>
> You are right that more background will benefit the understanding, as the different basis choices will be less known in the ML community. Please let us know if the following helps.
>
> Cartesian GTOs, as written in equation (16), are a standard choice of basis functions in quantum chemistry [1], implemented in most major quantum chemistry codes, such as PySCF (see the pyscf.gto package) and QChem (See the PURECART function of QCHEM). They are tightly related to spherical GTOs (we will elaborate on this in the answer to the next question). Both Cartesian and spherical GTOs mostly share the same benefits and challenges, with spherical GTOs being preferred recently due to higher numerical stability and compactness. The plane wave basis is preferred for solids but also used for molecules due to their simplicity, numerical stability, and absence of basis-set superposition errors.
>
> In the context of ML models, Cartesian basis functions are natural for the same reason spherical harmonics-based basis functions are natural: The known transformation of the angular part allows for the construction of rotation equivariant architectures. This is also the reason that many equivariant graph neural networks, e.g. TensorNet[2] and indeed HotPP[3] (the backbone for ELECTRA), use Cartesian tensors as the basis for the equivariant operations.
>
> The justification for our choice of using Gaussians is that the reduced complexity makes the model faster, while the connection to Cartesian basis functions allows us to enforce the symmetry properties.
>
> We will add a paragraph based on the discussion above to the paper to better motivate our choices.
>
> >**_... the connection between geometric tensor rank and quantum angular momentum is fundamentally different and requires clarification.
> Specifically, the relationship between Equation (15) and Equation (16) is ambiguous. There is no explanation of why the angular quantum number in both equations should be interpreted identically._**
>
> Thank you for the feedback. We agree that the link between tensor rank and angular momentum is less known in the ML community and will provide more background:
> We can write any solid spherical harmonic as a linear combination of monomials in the following way (see for example, [4], equation 2):
> $$
> r^lY_{l,m}(\theta, \phi) = \sum_{n_x + n_y + n_z = l} C_{n_x,n_y,n_z}^{l,m} x^{n_x} y^{n_y} z^{n_z}
> $$
> Similarly, the traceless symmetric component of the Cartesian tensor of rank $l$ can be spanned by solid harmonics of degree $l$.
>
> Therefore, the angular momentum quantum number $l$ and $n_x + n_y + n_z = l$ are indeed very related and also both commonly called "angular momentum" in the quantum chemistry literature.
>
> The equation above is unfortunately somewhat involved to show, so in our paper, we will refer the interested reader to [5].
>
> >**_Typos and Minor Issues:..._**
>
> Thank you for the in-depth feedback. We will address all notation issues and typos and make sure to re-check for any potential typos we missed!
>
> >**_Interpretation of Figures 1 and 2: What is the frame or vector space represented in these figures? Specifically, are the vectors shown in Figures 1a and 1c the predicted mean positions in Equation (6)? A detailed explanation of what each vector represents in these figures would greatly improve clarity._**
>
> Yes, these are the vector-valued ($l=1$) outputs which we use, after multiplication with predicted scalar factors, as our mean predictions in equation (6). We will add this to the caption.
>
> >**_GPU Memory Requirements: How much GPU memory was used during training and inference? Are there specific trade-offs in inference performance compared to prior work?_**
>
> Everything (both training and inference) runs on a single NVIDIA RTX 3090 with 24GB, with batch size 1, see line 285.  For comparison, SCDP was trained on 4xNVIDIA A100 80 GB with a batch size of 4. There are no specific trade-offs we are aware of; on our tests, ELECTRA is consistently both much faster and more accurate compared to competing methods.
>
>
> >**_Channel Dimension Limitation: Line 184 specifies $N_c = 8 \cdot M_e$. Does this imply that the model can only handle atoms with fewer than 8 valence electrons? How generalizable is the model to elements beyond the first two rows of the periodic table?_**
>
> This is not a fundamental limitation. $N_c$ is a hyperparameter, and $N_c = 8 \cdot M_e$ was just a heuristic choice to allocate more expressivity to atoms with more valence electrons. This is analogous to traditional basis sets allocating more basis functions to heavier atoms. However, nothing prevents us from breaking this rule of thumb. If we were to include new atom types, we would of course have to retrain the network using different parameters, but this is the same for any atomistic model, not just density prediction models.
>
> >**_Density Grid Sampling: How did you handle sampling of the density during training? Did you employ specific sampling algorithms over the reference density grids in the dataset?_**
>
> Thank you for the question! We did not sample. We will clarify this by adding the following to our paper:
> _"ELECTRA was trained on the standard QM9 benchmarking dataset using the full grid for both training and inference."_
>
>
> >**_Practical Use Cases: Could you elaborate on whether ELECTRA’s predicted electron densities can improve computational physics simulations, such as providing initial guesses for SCF iterations or being directly used in orbital-free DFT calculations?_**
>
> Yes, they can. A density given on a regular grid can be efficiently transformed into plane wave basis using fast Fourier transformation to initialize SCF cycles. This follows previous work, for example, ChargE3Net [4], and is also how plane wave codes conventionally initialize their calculations from SAD guesses. This is straightforward in practice; in the VASP code base, you can provide custom initializations provided on the grid using the ICHARG=1 setting. (we are not allowed to post links to the documentation, but it can easily be found)
>
> To demonstrate this point, we used the density predictions of ELECTRA to initialize DFT calculations for our QM9 test molecules and achieved an **average reduction of 50.72 %SCF cycles**  compared to conventional initialization. We will add these numbers and a better description of how this is done to our paper to demonstrate practicality!
>
> Orbital-free DFT provides additional motivation as it should work in principle, but we did not try it out in practice.
>
>
> ---
>
> We hope we were able to address all concerns. Since the primary concern arose from a misunderstanding due to notational oversight, we hope that the reviewer agrees that this and other issues have been resolved.
> In that regard, we would also like to add that we sincerely appreciate that the reviewer acknowledges the novelty and superior performance and efficiency of our approach. Thus, we hope that our responses have further reinforced this assessment and convinced the reviewer to revise their final recommendation accordingly.
> Please let us know if there are remaining problems or improvements that could be addressed before the paper should be accepted.
>
> ---
>
> [1] (Helgaker, Trygve, Poul Jørgensen, and Jeppe Olsen. 2000. Molecular Electronic-Structure Theory. John Wiley & Sons, Ltd. - Chichester. Section 9.2.1)
>
> [2] Simeon, G., & De Fabritiis, G. (2023). Tensornet: Cartesian tensor representations for efficient learning of molecular potentials. Advances in Neural Information Processing Systems, 36, 37334-37353.
>
> [3] Wang, J., Wang, Y., Zhang, H., Yang, Z., Liang, Z., Shi, J., ... & Sun, J. (2024). E (n)-Equivariant cartesian tensor message passing interatomic potential. Nature Communications, 15(1), 7607.
>
> [4] Koker, Teddy, et al. "Higher-order equivariant neural networks for charge density prediction in materials." npj Computational Materials 10.1 (2024): 161.
>
> [5] Ribaldone, Chiara, and Jacques Kontak Desmarais. "Spherical to Cartesian Coordinates Transformation for Solid Harmonics Revisited." arXiv preprint arXiv:2412.16733 (2024).

---

> ### Comment · Reviewer_9wy9 · 2025-08-05
>
> Thank you to the authors for carefully considering my comments and questions. First, I would like to commend the results on SCF cycle reduction—this is a remarkable contribution that significantly enhances the practical value of the work. I also appreciate the overall improvement in density estimation achieved by the proposed framework.
>
> However, I would like to clarify that some concepts are misused in the manuscript. Specifically, I find the use of the term invariant in the context of electron density somewhat awkward—even when conditioned on the molecular configuration. In group theory, invariance under a group action is typically defined as $f(Rx) = f(x)$. Given that the motivation and architecture of your work are grounded in handling symmetries, I recommend using group-theoretic terminology more deliberately and precisely. A more rigorous explanation—supported by clear definitions and formal justifications—of how symmetry is handled in your framework would substantially strengthen the paper.
>
> Regarding the use of Cartesian tensors, I appreciate the references you provided, which helped me better understand your intentions. However, I still feel that further explanation of how Cartesian tensors are used for representing electron density is needed in the manuscript. Additionally, there appears to be some confusion between geometrical tensor rank (ex. l=1 for force, l=2 for stress 3x3 matrice) and angular momentum quantum numbers, particularly in Equation (5) and lines 140–141. While I agree that using CGTOs (Cartesian Gaussian Type Orbital) makes sense, the proposed model “HotPP” does not appear to be a Cartesian tensor in the conventional sense, but rather a tensor used in mathematical physics—often referred to as a geometrical tensor. I strongly encourage the authors to improve the consistency and rigor of their mathematical formulations. Despite the strong empirical performance, such conceptual imprecisions may mislead readers and weaken the scientific impact of the work.
>
> Finally, regarding the reported reduction in SCF steps, to the best of my knowledge, this metric is sensitive to various hyperparameter choices—such as the functional, pseudopotentials, and the integration grid level. I hope that these choices were adequately and fairly reported in future manuscripts.
>
> Other aspects of the rebuttal addressed my concerns well. While I am hesitant to recommend outright acceptance at this stage, I believe this work is both timely and promising. With improved **mathematical rigor around equivariance**, **clearer distinction between geometric tensors and Cartesian basis functions**, **consistent notation**, and **careful correction of minor issues and typos**, this paper has the potential to make a significant contribution to the ML community. Accordingly, I have raised my score.

---

> > ### Author Response · Authors · 2025-08-06
> >
> > Thank you for the positive comments and constructive feedback!
> >
> > >First, I would like to commend the results on SCF cycle reduction—this is a remarkable contribution that significantly enhances the practical value of the work ... regarding the reported reduction in SCF steps, to the best of my knowledge, this metric is sensitive to various hyperparameter choices—such as the functional, pseudopotentials, and the integration grid level. I hope that these choices were adequately and fairly reported in future manuscripts.
> >
> > You are right that the SCF cycles are sensitive to the hyperparameter choices. We therefore use the exact same parameters that the dataset authors used to create the QM9 dataset. We will add the settings to the appendix for convenience.
> >
> >
> > >However, I would like to clarify that some concepts are misused in the manuscript. Specifically, I find the use of the term invariant in the context of electron density somewhat awkward—even when conditioned on the molecular configuration.
> >
> > We will add the group theoretically rigorous notation $(\mathcal{T}_R \cdot \rho ) (r) = \rho(Rr)$ that you mentioned to our paper. Could you send the reference again? It seems it got lost in the original review. We think the viewpoint of invariance under simultaneous transformation of nuclei and electron as written above is mathematically correct and also valuable, though, as it is less abstract and describes how one can, for example, implement a unit test in a practical code implementation. We therefore think both views complement each other and aid understanding. Would you agree with this?
> >
> >
> > >However, I still feel that further explanation of how Cartesian tensors are used for representing electron density is needed in the manuscript.
> >
> > The neural network's outputs are Cartesian tensors of rank 0, 1, and 2, which we respectively use as the weights, position, and covariance matrices of the Gaussians in the Gaussian mixture model that represent the electron density. As shown in our Appendix B, the transformation properties of Cartesian tensors (equation 5) lead to a Gaussian mixture model/electron density that transforms correctly under simultaneous rotation of electron and nuclei.
> > Is a section like this what you had in mind, or is there another aspect you would like to see explained more?
> >
> >
> > >Additionally, there appears to be some confusion between geometrical tensor rank (ex. l=1 for force, l=2 for stress 3x3 matrice) and angular momentum quantum numbers, particularly in Equation (5) and lines 140–141.
> >
> > We will replace the $l$ in equation (5) with an $r$ to avoid confusion.
> >
> >
> > >... the proposed model “HotPP” does not appear to be a Cartesian tensor in the conventional sense, but rather a tensor used in mathematical physics—often referred to as a geometrical tensor ...
> >
> > The papers for both HotPP [1] and the related TensorNet [2] refer to the features and outputs in their models specifically as cartesian tensors, characterized by their transformation behavior (equation (5) in our paper or equivalently equation (1) in [1]). We therefore think "cartesian tensor" is the right terminology for HotPP's outputs. Could you explain why you think this is not correct?
> >
> >
> > >With improved mathematical rigor around equivariance, clearer distinction between geometric tensors and Cartesian basis functions, consistent notation, and careful correction of minor issues and typos, this paper has the potential to make a significant contribution to the ML community. Accordingly, I have raised my score.
> >
> > In our revised version, we will make sure to carefully double-check the notational consistency and typos. We will also add a section about group theoretical background similar to the one in the appendix of [3].
> >
> > To avoid confusion between "cartesian basis function" and "cartesian tensors" we will point out that the outputs of the neural network are "cartesian tensors", which we use as weights/gaussian positions/covariances and "cartesian basis functions" refer to the monomials of the x,y,z components of the displacement vector between grid point position and gaussian position.
> >
> >
> >
> > [1] Wang, Junjie, et al. "E (n)-Equivariant cartesian tensor message passing interatomic potential." Nature Communications 15.1 (2024): 7607.
> >
> > [2] Simeon, Guillem, and Gianni De Fabritiis. "Tensornet: Cartesian tensor representations for efficient learning of molecular potentials." Advances in Neural Information Processing Systems 36 (2023): 37334-37353.
> >
> > [3] Liao, Yi-Lun, et al. "Equiformerv2: Improved equivariant transformer for scaling to higher-degree representations." arXiv preprint arXiv:2306.12059 (2023).

---

> > > ### Comment · Reviewer_9wy9 · 2025-08-07
> > >
> > > Thank you for taking my comments and for engaging in a constructive discussion throughout.
> > >
> > > First, regarding the group-theoretical aspect, I have also spent time thinking about how best to rigorously express the symmetry properties of electron density, and I found that simply stating it as “invariant” or “equivariant” under group actions may not be sufficiently precise. Formally, a function f: X \to Y is invariant under a group action g \in G if:
> > >
> > > $$f(g \cdot x) = f(x)$$
> > >
> > > (This concept was firstly used in your initial manuscripts which was incorrect.)
> > >
> > > However, in the case of electron density, the transformation is not solely acting on the input space. Rather, the group action effectively transforms the *function itself*, which aligns more closely with the concept of the **regular representation**, where the transformation is expressed as:
> > >
> > > $$f(g \cdot x) = (T_{g^{-1}} f)(x)$$
> > >
> > > This interpretation is conceptually related to your statement about the *“invariance under simultaneous rotation of nuclei and electrons.”* While your revised notation is not technically incorrect, it is somewhat unconventional. I would therefore recommend expressing it more explicitly using the language of group representations for clarity and rigor.
> > >
> > > I had previously shared a reference related to this point, but it appears that it was not visible to the authors. For completeness, I am including the references again below, which I believe will help support the conceptual clarification I am suggesting.
> > >
> > > [1]: Bekkers, Erik J., et al. "Roto-translation covariant convolutional networks for medical image analysis." Medical Image Computing and Computer Assisted Intervention–MICCAI 2018: 21st International Conference, Granada, Spain, September 16-20, 2018, Proceedings, Part I. Springer International Publishing, 2018.
> > >
> > > [2] Bekkers, Erik J. "B-spline cnns on lie groups." *arXiv preprint arXiv:1909.12057* (2019).
> > >
> > > [3] Brandstetter, Johannes, et al. "Geometric and physical quantities improve e (3) equivariant message passing." arXiv preprint arXiv:2110.02905 (2021).
> > >
> > > [4] Finzi, Marc, et al. "Generalizing convolutional neural networks for equivariance to lie groups on arbitrary continuous data." *International conference on machine learning*. PMLR, 2020.
> > >
> > > ---
> > >
> > > Second, regarding the use of Cartesian tensors, I acknowledge that my earlier statement—that HotPP is not based on Cartesian tensors—was inaccurate. I was confused and had difficult in distinction with the Cartesian tensors and Cartesian basis functions at first. Initially, I understood the manuscript as employing Cartesian tensor–based networks to better represent Cartesian basis functions since both architectures are aligned with the concepts of Cartesian. However, upon further reading, I recognize this was a misunderstanding. This is no longer a concern for me.
> > >
> > > Thank you again for your thoughtful and detailed responses. This discussion has significantly clarified the intent and contributions of the work to me. Although I had reservations based on the initial version of the manuscript, I now believe that this work is suitable for presentation at the conference and will be valuable to the broader machine learning community. I will accordingly update my score, clarity and soundness based on the discussion and fixed manuscripts.

---

> > > > ### Author Response · Authors · 2025-08-08
> > > >
> > > > Thank you for the provided references; you are right, in group theoretical language, a more standard way to express the transformation behavior of the electron density is to say that it transforms under rotation with the left-regular representation of the rotation. We will make this formulation as precise as possible and add it to our manuscript!
> > > >
> > > > We are glad we were able to clear up all your concerns!
> > > >
> > > > Thank you again for the active and constructive discussion; your comments helped us a lot in strengthening the clarity and precision of our work.

---

### Official Review · Reviewer_qQZb · 2025-07-02

**Clarity:** 3
**Significance:** 3
**Originality:** 3
**Rating:** 4
**Confidence:** 4

**Summary:**

The paper introduces ELECTRA, a model for predicting electronic charge density using floating orbitals, a concept rooted in quantum chemistry but adapted in a data-driven manner via an Equivariant backbone neural network. Incorporates a symmetry-breaking mechanism using eigenvectors of the moment of inertia tensor to enhance expressivity while preserving rotational equivariance, addressing limitations of highly symmetric inputs. Compared to prior work (e.g., Fu et al., 2024; Rackers et al., 2023), ELECTRA’s use of floating orbitals and symmetry-breaking is distinct, as it avoids reliance on fixed basis sets or bond-centered orbitals, offering a fully data-driven solution (Page 9). This is a significant departure from traditional DFT methods and other ML-based approaches like SCDP or Charge3Net, which often use atom-centered or spherical harmonic representations.

**Questions:**

Can you also address some examples of molecules where this approach fails or has less advantages?

**Ethical Concerns:**

["NO or VERY MINOR ethics concerns only"]

**Quality:**

3

**Strengths And Weaknesses:**

Strengths - Incorporating floating orbitals with an Equivariant backbone neural network is novel.
Weaknesses - The author should also address when this would fail in certain cases.

---

> ### Author Rebuttal · Authors · 2025-07-29
>
> We thank the reviewer for recognizing the novelty and strong performance of ELECTRA. Below, we address the question of whether our model struggles with some molecules.
>
> >**_"Can you also address some examples of molecules where this approach fails or has less advantages?"_**
>
> Thank you for the question!
> We were looking through the samples of the QM9 test dataset with the largest errors. The highest NMAE error is 1.43 %. For comparison, the conventional SAD (superposition of atomic densities) guess, which DFT is usually initialized with, is around 13% on average. The molecule (C3H5N3O2) is a significant outlier containing an Oxadiazole-type structure and an NH3 side group. These two motifs have heavily distorted symmetry in the charge density and are rare and underrepresented in the data. The error then drops quickly for the next worst molecules. We did not find any particular pattern between these bad performing molecules apart from having some motives that are rare in the train set.
> Generally, the error is higher around heavier atoms, which is also expected, as there are fewer heavy atoms in the train set, and the electron density is more complex around them. Overall, the model seems robust to most molecules in the QM9 dataset, which tests for molecule generalization, and the MD dataset, which tests generalization to conformal changes.
>
> Some current limitations of our model from a practical point are that it is only trained on neutral, organic molecules. Expanding this to more complicated settings, such as charged systems or highly correlated and diffuse systems, will require extra work. We see this as orthogonal to the subject of our paper, which tries to fundamentally explore charge density representation using floating orbitals. There is nothing in principle preventing us from training on different kinds of systems (and the same limitations hold for competing published methods as well).
>
> One potential downside of our approach specifically, is that chemists have built up a lot of intuition to reason about spherical harmonics-based orbitals. Our gaussians are less interpretable, which can be a downside in some situations.
>
> We hope this answers your question. If not, can you tell us more about the kind of disadvantages you were thinking about?
>
>
> ---
>
> More generally, could you please let us know if there are parts of the paper that are still unclear or would benefit from more polishing to warrant an increased score? Considering that our model beats the current state-of-the-art, which was published at last year's Neurips by wide margins (10x faster with better accuracy and trained with less memory), we do not understand where we can improve more without more detailed feedback.

---

> > ### Author Response · Authors · 2025-08-05
> >
> > Dear Reviewer qQZb,
> >
> > Thank you again for your question about the limitations of our model.
> >
> > As a brief follow-up directly related to your question, we examined how lower-accuracy molecules affect a practical use case.  In DFT workflows, a primary use of ML densities is to provide better initial densities that accelerate SCF convergence and reduce compute. Using our existing setup, results show that initializing SCF with densities predicted by ELECTRA reduces the number of SCF iterations by 50.72% on average relative to default initialization.
> >
> > Consistent with our NMAE analysis posted as an answer to your original question, the smallest reduction in SCF steps ( around 17%) occurs for the highest-error molecule (C3H5N3O2, mentioned above), which, as mentioned, contains very rare and unusual motifs. However, ELECTRA yields significantly larger savings on most molecules (several SCF step reductions of more than 60%, with a mean of 50.72%). This indicates that lower density error translates into larger SCF savings (and vice versa), which is a useful sanity check on the model. We will add a description of these results in the revision.
> >
> > We hope this fully addresses your question and that the additional validation informs your final assessment and recommendation.

---

> > ### Comment · Reviewer_qQZb · 2025-08-07
> >
> > thanks the authors for the clarification!

---

### Note · Authors · 2025-08-11

Dear Reviewers and AC

Thank you for your helpful and encouraging discussions.

We are happy that all reviewers acknowledged the novelty and state-of-the-art performance and efficiency of our work.
We appreciate that we were able to collaborate with the reviewers in resolving any major concerns, and that this resulted in all reviewers converging on recommending our paper for the conference.

We will incorporate all changes requested by each reviewer into the revised version of our paper, as detailed in the individual discussions.
Below, we shortly summarize the main points that came out of the discussions:

**All reviewers**
To demonstrate practical utility, we will add the details on the SCF-initialization experiment, which shows that ELECTRA reduces SCF iterations by 50.72% on average relative to default initialization. We will document the SCF-initialization protocol and exact QM9 DFT settings.

**Reviewer qQZb**
We will explicitly discuss any limitations of the model on specific molecules.

**Reviewer 9wy9**
We will correct Eq. (4) and adopt the standard group-theoretic phrasing that the density transforms under rotations via the (left-)regular representation. We will add a background section on the necessary group theory and check our work for any weaknesses in clarity and notational rigour, as well as for typos.  We will also add clarifications wherever necessary to distinguish Cartesian tensors and Cartesian basis, and also add the discussed details regarding our training protocol.

**Reviewer amF4**
We will include the mathematical argument for why displacement-based message passing induces a persistent bias direction, and support this analysis with the discussed extra plot. We will also briefly discuss our backbone exploration and emphasize that Gaussian splatting was only a loose inspiration for what the Reviewer correctly described as a (equivariant) GMM-like spatial function approximation.

**Reviewer 5buA**
We will add the "overfit GMM" experiment (same number of Gaussians as ELECTRA), showing that the choice of a Gaussian basis is expressive enough to capture the electron density. We will note that adding ReLU for positivity did not change outcomes empirically.
We will add the atomic embedding details and include the n-hop argument in the local symmetry proof in the appendix.


Thank you again for the constructive process and consideration.

---

### Decision · Program_Chairs · 2025-09-17

**Decision:**

Accept (spotlight)

**Comment:**

This paper introduces a model for electron charge density prediction that leverages floating basis functions, i.e., Gaussian mixture with non-linearities. The approach demonstrates significant improvements in both predictive accuracy and computational efficiency. In particular, the paper demonstrates practical utility through reductions in SCF iterations .

The reviewers generally agreed on the novelty and strength of the contributions, especially the data-driven treatment of floating orbitals and the demonstrated efficiency. While some concerns were raised about conceptual precision in symmetry arguments and notational inconsistencies, the authors addressed these points during the rebuttal.

Overall, I believe the paper is timely and impactful, hence recommend acceptance.